# Comparative transcriptomics reveals highly conserved regional programs between porcine and human colonic enteric nervous system

Tao Li [1], Marco Morselli [2], Trent Su[3], Mulugeta Million[1], Muriel Larauche[1], Matteo Pellegrini[2], Yvette Taché[1,4] & Pu-Qing Yuan [1,4 ✉]

The porcine gut is increasingly regarded as a useful translational model. The enteric nervous system in the colon coordinates diverse functions. However, knowledge of the molecular profiling of porcine enteric nerve system and its similarity to that of human is still lacking. We identified the distinct transcriptional programs associated with functional characteristics between inner submucosal and myenteric ganglia in porcine proximal and distal colon using bulk RNA and single-cell RNA sequencing. Comparative transcriptomics of myenteric ganglia in corresponding colonic regions of pig and human revealed highly conserved programs in porcine proximal and distal colon, which explained >96% of their transcriptomic responses to vagal nerve stimulation, suggesting that porcine proximal and distal colon could serve as predictors in translational studies. The conserved programs specific for inflammatory modulation were displayed in pigs with vagal nerve stimulation. This study provides a valuable transcriptomic resource for understanding of human colonic functions and neuromodulation using porcine model.

[1] CURE/Digestive Diseases Research Center, Vatche and Tamar Manoukian Digestive Diseases Division, Department of Medicine, David Geffen School of Medicine, University of California at Los Angeles (UCLA), Los Angeles, USA. [2] Department of Molecular, Cell, & Developmental Biology, UCLA, Los Angeles, USA. [3] Department of Biological Chemistry, UCLA, Los Angeles, USA. [4] VA Greater Los Angeles Healthcare System, Los Angeles, USA. ✉email: pqyuan@mednet.ucla.edu

The porcine gastrointestinal (GI) tract is increasingly regarded as a useful translational research model due to its structural and functional similarities with human. Compared to mouse, rat, dog, cat, or horse, both pig and human are omnivores and colon fermenters, and have similar metabolic and intestinal physiological processes and microbial composition[1–4]. Among important anatomical similarities relevant to colonic function, both taenia and haustra/sacculation are present in porcine and human colon, while missing in dogs and most rodents. However, it is to note that unlike in human, the porcine proximal colon is orientated in a spiral fashion[2,5]. Additionally, compared with human, pig has a similar chromosomal structure and a comparably sized genome with a 60% sequence homology and 93% correspondence in relevant biomarkers[6–8]. The porcine size is also relevant for closely mimicking drug dose volumes used in human and for evaluating medical devices[9]. Thus, the porcine model has untapped potential for a greater understanding of the underlying colonic functions of human in health and diseases[10].

The GI tract is highly innervated both by extrinsic sympathetic, parasympathetic and visceral afferent neurons originating outside the GI tract, and by an intrinsic nervous system, the enteric nervous system (ENS). The ENS is endowed with complex reflex circuits that control a variety of GI physiological functions via networks of neurons and glia in the mucosal and muscular layers[11]. Uniquely the ENS is the only part of the peripheral nervous system that can function independently of the central nervous system. Therefore, an intact ENS is essential for life and its dysfunction is often linked to digestive disorders[11,12]. In mammalian colon, the ENS comprises two main ganglionated plexuses: myenteric plexus, located between longitudinal and circular layers of muscularis externa, and submucosal plexus within the submucosa. Unlike the submucosal plexus in rodents that is single-layered[13], in both porcine and human colon, the submucosal plexus consists of an inner submucosal plexus, located close to muscularis mucosae, and an outer submucosal plexus, located on the luminal side of the circular muscle layer[14]. Despite the anatomical similarity, it is still unknown to what extent transcriptional programs are conserved in the colonic ENS between pig and human. Such information is of importance for the use of porcine model in translational studies.

Recent studies in mice have demonstrated heterogeneity in neuronal identity, morphology, projection orientation and synaptic complexity within the ENS located in different colonic segments in line with the diversity of colonic motility patterns[15]. The region-dependent molecular characterization of porcine ENS has been little investigated so far. In addition, colonic functions are modulated through parasympathetic innervation, responsible for regulating colonic secretion and motility[16]. We have recently reported that electrical vagal nerve stimulation (VNS) triggered pan-colonic contractions in the pig[17]. However, it is unknown how such VNS influences colonic transcriptional programs and whether the impacts of VNS are region-specific.

The incomplete molecular characterization of porcine ENS is largely due to longstanding technical challenges in isolating enteric neurons[18]. Laser-capture microdissection (LCM) has been the method of choice for accurately targeting and capturing cells of interest from a heterogeneous tissue sample[19] like the colon[20]. The enriched cells can be used for bulk RNA sequencing (RNA-seq), which becomes a widely adopted method for profiling transcriptomic variations[21] and is a promising choice to characterize the region-dependent molecular profiling in ENS. More recently, single-cell RNA sequencing (scRNA-seq) has emerged as a powerful method for exploring gene expression profile at single-cell resolution[22]. However, its high cost limits its application across a large population of samples[23]. Considering this constraint, several computational methods have been exploited to deconvolve cell type composition in bulk RNA-seq data using scRNA-seq reference datasets[24–27]. This has largely facilitated the use of scRNA-seq data from a small number of subjects in clinical setting involving a large number of subjects.

In the present study, we profiled the transcriptomes of myenteric and inner submucosal ganglia (MG, ISG) from porcine proximal, transverse and distal colon (p-pC, p-tC, p-dC), and MG in parallel from human ascending, transverse and descending colon (h-aC, h-tC, h-dC) using LCM coupled with bulk RNA-seq analysis. By cross-comparison of the transcriptional profiling of MG in the corresponding regions between porcine and human colon, we aimed to characterize the conservation of gene programs derived from orthologous genes. Then, based on the high-quality orthologous genes, we evaluated discrepancy of the region-specific gene programs in MG and ISG after electrical stimulation of the celiac branch of the abdominal vagus nerve in the anesthetized pigs. Additionally, we performed scRNA-seq coupled with bulk gene expression deconvolution to identify cell-type marker genes and putative interactions among neurons and glia in MG of porcine colon and further revealed the regional transcriptomic heterogeneity in the ENS of the pigs with and without VNS (Fig. 1).

## Results

**Transcriptomic similarities of colonic myenteric ganglia between pig and human.** We cross-compared the corresponding regional transcriptional profiling of MG between porcine and human colon to determine to what extent the porcine model offers access to the colonic translational research based on the transcriptomic similarities. A possible explanation for those similarities was gleaned from comparisons of empirical cumulative density function (ECDF) profiles of the averaging normalized transcription levels of each high-quality orthologous gene. Both Kolmogorov-Smirnov and Pearson's Chi-squared test were applied to quantify the distance between ECDF profiles. Results demonstrated high similarities of probability distributions of human-porcine regional gene expression profiling in MG between corresponding colonic segments in the pig and human (p-pC vs h-aC, p-tC vs h-tC, and p-dC vs h-dC). This was supported by each pair of ECDF plots being almost merged and the lack of significant difference between the pair of ECDF profiles ($p > 0.05$), assessed by Kolmogorov-Smirnov and Pearson's Chi-squared test (Fig. 2a, c, Supplementary Data 1). Thus, the human colonic segments (h-aC, h-tC, h-dC) could be predicted according to that of corresponding porcine colonic segments. In addition, pair-wise comparison showed a significant Spearman's correlation in the corresponding regions between the two species ($p < 0.001$) (Fig. 2b), suggesting the transcriptomic similarities of analogous MG in the corresponding colonic segments between pig and human.

The lists of differentially expressed genes (DEG) were derived from comparisons of MG between analogous colonic regions across species and between ISG and MG in porcine colonic segments. The DEG lists were used to create the enrichment maps of Gene Ontology Biological Processes (GOBP) terms with a false discovery rate (FDR)-adjusted p value threshold of $q < 0.05$. The functional linkages related to the orthologous DEG were extracted to reflect functional similarities across species. Briefly, the orthologous DEG were searched in the grouped networks of GOBP terms and all terms containing the DEG was highlighted. Consequently, the information for the DEG-enriched terms and the number of genes shared by the connected terms (functional linkages) was used for comparisons of functional differences. The results showed that there were 12 porcine orthologous DEG when comparing h-aC-MG and h-dC-MG, of which 9 were upregulated

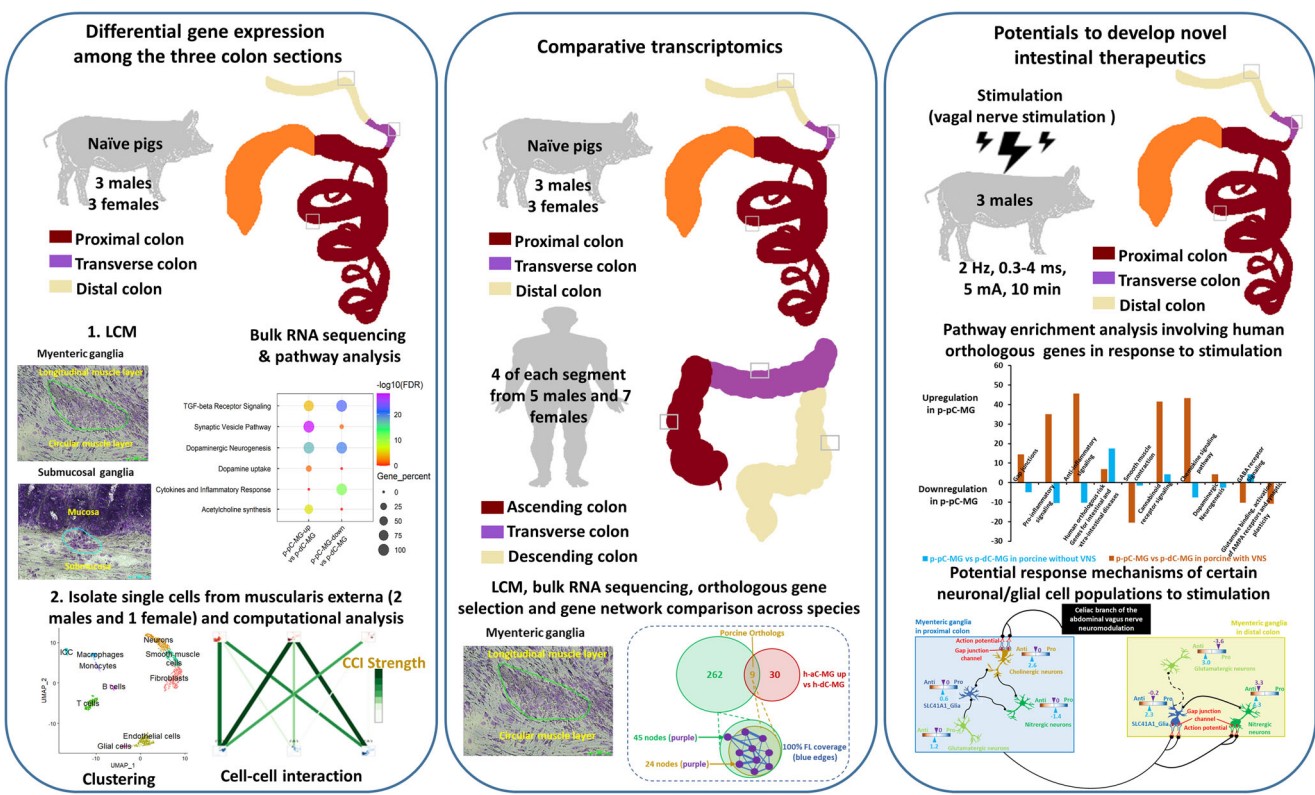

**Fig. 1 Experimental design in the study.** The transcriptomes were profiled in myenteric and inner submucosal ganglia (MG, ISG) from proximal, transverse and distal colon (p-pC, p-tC, p-dC) of six naive pigs, and MG in parallel from ascending, transverse and descending colon (h-aC, h-tC, h-dC, 4 of each) from 12 patients (5 males and 7 females) using laser-capture microdissection (LCM) coupled with bulk RNA-seq analysis. Gene homology search was performed to obtain the orthologous genes between human and pig. The high-quality orthologous differentially expressed genes (DEG) derived from comparisons of MG between analogous colonic regions across species were extracted and the coverage of their functional linkages in the gene networks was used to reflect functional similarities across species. Based on the orthologous genes, the discrepancy of the region-specific gene programs was evaluated in MG and ISG after electrical stimulation of the celiac branch of the abdominal vagus nerve in the anesthetized pigs. Additionally, single-cell RNA-seq (scRNA-seq) was performed on the muscularis externa containing myenteric ganglia in three naive pigs (2 males and 1 female). Approximately, an average of 3200 cells (1400/3000/5300) from each p-pC individual, 3800 cells (2800/2200/6300) from each p-tC individual and 4100 cells (2400/5000/4920) from each p-dC individual were loaded on the 10× Genomics Chromium platform, generating 9 scRNA-seq datasets. This yielded an average of 1930 (735/2091/2965), 2318 (1993/1903/3057) and 2559 (1381/3918/2378) mapped single cells in each of p-pC, p-tC, and p-dC, respectively. The scRNA-seq was coupled with bulk gene expression deconvolution to identify cell-type marker genes and putative interactions among neurons and glia in MG of porcine colon and further revealed the regional transcriptomic heterogeneity in the enteric nervous system of the pigs with and without VNS.

and 3 were downregulated. However, there were 483 DEG in comparison of p-pC-MG and p-dC-MG, of which 271 were upregulated and 212 were downregulated (Fig. 2d). The 271 and 212 DEG generated 4176 and 5897 functional linkages, respectively. These 12 porcine orthologous DEG were then searched in the grouped networks of Biological Processes (BPs) (nodes shown in Fig. 2d), mediated by the 483 DEG. Although the orthologous DEG were directly involved in around 64% ($\frac{24+39}{45+51}*100$) of the grouped BP networks, the functional linkages related to the orthologous DEG achieved 100% linkage coverage. The functional linkages related to all DEG in comparison of h-aC-MG and h-dC-MG were the same as those related to the orthologous DEG (Fig. 2d). Moreover, these conserved functional linkages accounted for more than 96% of all functional linkages of the regulatory networks responding to VNS when comparing p-pC-MG and p-dC-MG. However, the 12 porcine orthologous DEG were directly involved in only about 43% ($\frac{47+48}{135+86}*100$) of the grouped BP networks under VNS. The partial coverage was shown when matching the functional linkages derived from human orthologous DEG in comparison of p-pC-MG and p-dC-MG and DEG in comparison of h-aC-MG and h-dC-MG

(Fig. 2d). The DEG were matched to the high-quality orthologous gene list, but we did not find any porcine orthologous DEG when comparing h-aC-MG or h-dC-MG with h-tC-MG. Therefore, only p-pC and p-dC were considered in the subsequent analyses.

**Preferential enrichment for genes associated with functions of myenteric and inner submucosal ganglia (MG, ISG) in porcine proximal or distal colon (p-pC, p-dC).** The functional linkages related to each DEG list in our study and those related to the DEG list screened by the high-quality human orthologous genes achieved almost 100% linkage coverage. The cell-subtype gene markers also fell into the high-quality orthologous gene list. Two databases (Gene Ontology Biological Processes and WikiPathways) were used to interpret the RNA-seq data. As mentioned above, the enrichment maps of GOBP terms were first created using the DEG lists. The DEG involved in the WikiPathways and Biological Processes (BPs) were searched in the enrichment maps. The GOBP terms were shared between different pathways (i.e. there were the DEG involved in the same GOBP terms), which was defined as the interactions between the networks mediated by

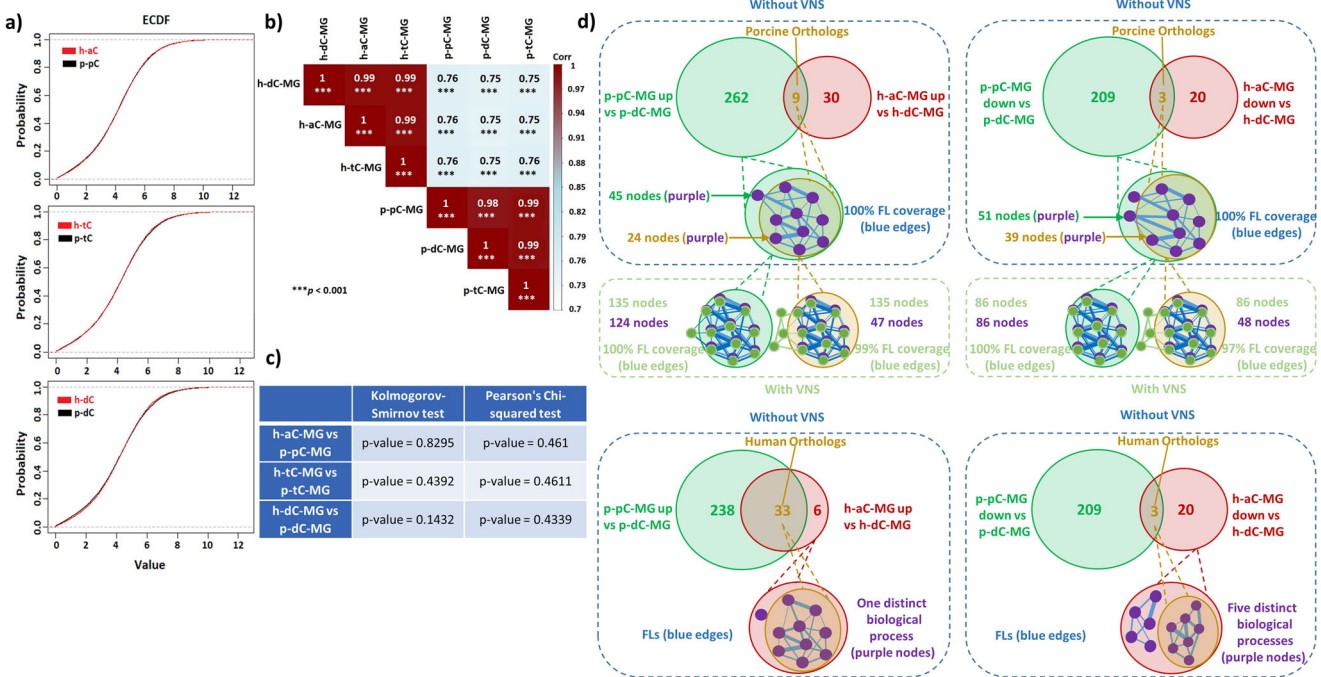

**Fig. 2 Transcriptomic similarities of colonic myenteric ganglia between pig and human.** Probability distributions of human-porcine regional datasets selected from the high-quality orthologous gene list between myenteric ganglia (MG) from porcine proximal, transverse and distal colon (p-pC, p-tC, p-dC) and in parallel from human ascending, transverse and descending colon (h-aC, h-tC, h-dC) **a**, Spearman's Correlation **b**, and evaluation of goodness of fit **c**. Color in **b** represents a correlation distribution from big (dark red) to small (light blue). Cartoon diagram showing regional differentially expressed gene (DEG) network matching between human and pig **d**. Gene Ontology Biological Processes terms (pathways) are shown as circles (nodes) that are connected with lines (edges) if the terms share many genes. Nodes are colored in purple (without VNS) and in light green (with VNS), and edges are sized on the basis of the number of genes shared by the connected pathways. *FLs* functional linkages, *VNS* vagal nerve stimulation, *Up* upregulated expression, *Down* downregulated expression.

the DEG. We mainly focused on interactions between the pathways of interest unless otherwise stated.

Similarities between p-pC and p-dC were found when comparing respective MG and ISG. The upregulated genes in MG of both p-pC and p-dC accounted for a larger percentage of identified genes involved in the top five BPs and led to the higher enrichment of the BPs (Supplementary Fig. 1). The BPs connected with those such as nerve development, neuronal projection guidance and neuron/glial cell differentiation. In addition, these genes in MG contributed to higher enrichment scores of the WikiPathways such as oncostatin M signaling pathway, glutamate binding, activation of AMPA receptors and synaptic plasticity, chemokine signaling pathway, effects of nitric oxide (NOS1), dopaminergic neurogenesis and TGF-beta receptor signaling. The downregulated genes in MG of both p-pC and p-dC made contribution to higher enrichment scores of the WikiPathways such as activation of angiotensin pathway, acetylcholine synthesis, synaptic vesicle pathway, neurotransmitter release cycle and dopamine transport and secretion (Fig. 3a, b).

However, there were still some differences found in comparison of MG and ISG between p-pC and p-dC. The upregulated genes in p-pC-MG contributed to higher enrichment scores of the WikiPathways such as PI3K-Akt signaling pathway, while the upregulated genes in p-dC-MG exclusively resulted in neurotransmitter uptake and metabolism in glial cells (Fig. 3a, b).

**Pathway enrichment changes in myenteric or inner submucosal ganglia (MG or ISG) in the proximal-distal axis of the porcine colon.** In p-pC-MG, the downregulated genes contributed to larger enrichment scores of the BPs (Supplementary

Fig 2a, c), which related to ENS development, blood vessel development and glial cell differentiation, while the BPs involving the upregulated genes were connected with ENS development, synaptic organization and chemical synaptic transmission. In ISG, the enrichment differences between p-pC and p-dC were not as large as those between p-pC-MG and p-dC-MG. The biggest enrichment disparity happened to blood vessel development and the downregulated genes in p-pC-ISG contributed to its higher enrichment score (Supplementary Figs 2b, d).

The upregulated genes in p-pC-MG led to the larger enrichment scores of the WikiPathways such as acetylcholine synthesis, synaptic vesicle pathway, neurotransmitter receptors and postsynaptic signal transmission, neurotransmitter release cycle and dopamine uptake, unlike those related to Oncostatin M signaling pathway, cytokines and inflammatory response, chemokine signaling pathway, NO/cGMP/PKG mediated neuroprotection, dopaminergic neurogenesis, TGF-beta receptor signaling and smooth muscle contraction (Fig. 3c). All listed pathways interacted with smooth muscle contraction. The upregulated genes in p-pC-ISG contributed to most prominent enrichment scores of nicotine activity on dopaminergic neurons, neurotransmitter receptors and postsynaptic signal transmission, dopaminergic neurogenesis, and dopamine biosynthesis process. The downregulated genes in p-pC-ISG resulted in enrichment scores of nitric oxide biological action, chemokine signaling pathway and TGF-beta receptor signaling (Fig. 3d).

**Distinct cell-type-specific pathway enrichment across porcine colonic segments detected by scRNA-seq.** The scRNA-seq data showed that there were 10 cell clusters in the muscularis externa of p-pC and p-dC detected with similar cell markers (Fig. 4a, b,

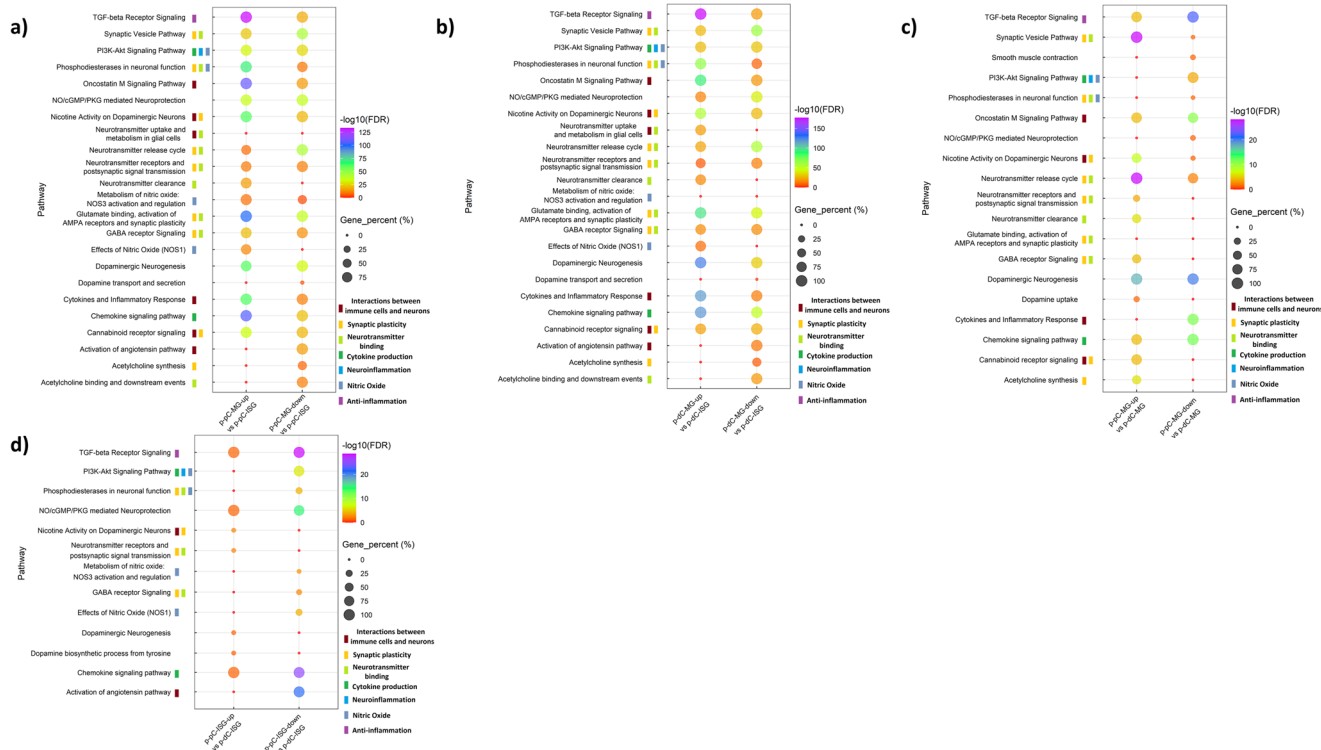

**Fig. 3 Preferential enrichment for genes associated with functions of myenteric and inner submucosal ganglia (MG, ISG) in porcine proximal or distal colon (p-pC, p-dC).** Comparison of pathway enrichment between MG and ISG in p-dC or p-dC **a**, **b**, and in MG or in ISG between p-pC and p-dC **c**, **d**. The bubble plots show the gene percentage and enrichment of WikiPathways. Circle size represents the ratio of the number of pathway-specific differentially expressed genes (DEG) and the number of total DEG in each DEG list (Gene_percent). Color represents a -log10(FDR) distribution from big (orange) to small (purple). The color bars mean the specified WikiPathways classified into designated clusters. Up, upregulated expression. Down, downregulated expression.

Supplementary Fig. 3). The neuronal and glial clusters were annotated with markers *GAP43* and *CLDN11*, respectively, which were validated by single-molecule fluorescence hybridization (smFISH). *GAP43* and *CLDN11* were co-expressed with *ELAVL4* and *GFAP*, respectively (Fig. 5a–c, m–o). Cell–cell interaction analysis revealed the stronger neuron-glia interaction in p-pC than p-dC (Fig. 4a, b). Three neuronal and two glial sub-clusters were further annotated with markers *ACLY* for cholinergic neurons, *GLS* for glutamatergic neurons, *ARHGAP18* for nitrergic neurons, *SLC41A1* and *SPC24* for two glial cell types and validated by smFISH with *ChAT*, *VGLU2*, *NOS1* and *GFAP*, respectively (Figs. 4c, d, 5d–l, p–s). Cell–cell interactions were also detected from ligand- to receptor-expressing neuronal and glial subsets including interactions between cholinergic neurons and nitrergic neurons or SLC41A1_ or SPC24_glia, glutamatergic neurons and cholinergic neurons or SLC41A1_glia, nitrergic neurons and SLC41A1_ or SPC24_glia, and SLC41A1_glia and SPC24_glia (Fig. 4e, Supplementary Table 1).

The resulting cell-type gene markers were then matched to the DEG lists from bulk RNA-seq data analysis. Consequently, the created cell-type DEG lists were used for pathway enrichment analysis. The WikiPathways with top ranking enrichment scores in comparison of p-pC-MG and p-dC-MG were selected to evaluate the contribution of each cell subset to pathway enrichment. Among the pathways involving the upregulated genes in p-pC-MG, cholinergic neurons played a leading role and contributed to enrichment of neurotransmitter release cycle, synaptic vesicle pathway, chemokine signaling pathway and TGF-beta receptor signaling (Fig. 6). Enrichment of neurotransmitter release cycle and synaptic vesicle pathway was also attributed to glutamatergic neurons and SLC41A1_glia, respectively. Among

the pathways involving the downregulated genes in p-pC-MG, nitrergic neurons contributed to the enrichment of dopaminergic neurogenesis, TGF-beta receptor signaling and chemokine signaling pathway. However, glutamatergic neurons contributed more to the enrichment of chemokine signaling pathway. Although cholinergic neurons contributed to the enrichment of neurotransmitter release cycle, their contribution was much less than that to enrichment of the pathway involving the upregulated genes in p-pC-MG (Fig. 6).

**Region-specific effects of vagal nerve stimulation on pathway enrichment in porcine proximal myenteric and inner submucosal ganglia (MG, ISG) and in porcine proximal and distal MG.** The enrichment ratio of the same pathway involving upregulated and downregulated genes in each DEG list (e.g., p-pC-MG versus p-dC-MG, or p-pC-MG versus p-pC-ISG) was calculated and compared with and without VNS in pigs to evaluate its influences at both single-cell and bulk resolution. The DEG, derived from comparison of p-dC-MG and p-dC-ISG with or without VNS, were involved in almost the same WikiPathways, despite different numbers of DEG. Only one differentially regulated gene under VNS was found, but enrichment of the Wiki-Pathway mediated by the gene was not significant (Supplementary Fig 5a), suggesting a significant overlap in the transcriptional landscape under these two situations. Similarly, comparison of p-pC-ISG and p-dC-ISG revealed that only small differences were observed with and without VNS (Supplementary Fig 5b).

The comparison without VNS showed that BP enrichment patterns were almost the same as with VNS based on comparison of p-pC-MG and p-pC-ISG. There were higher enrichment scores

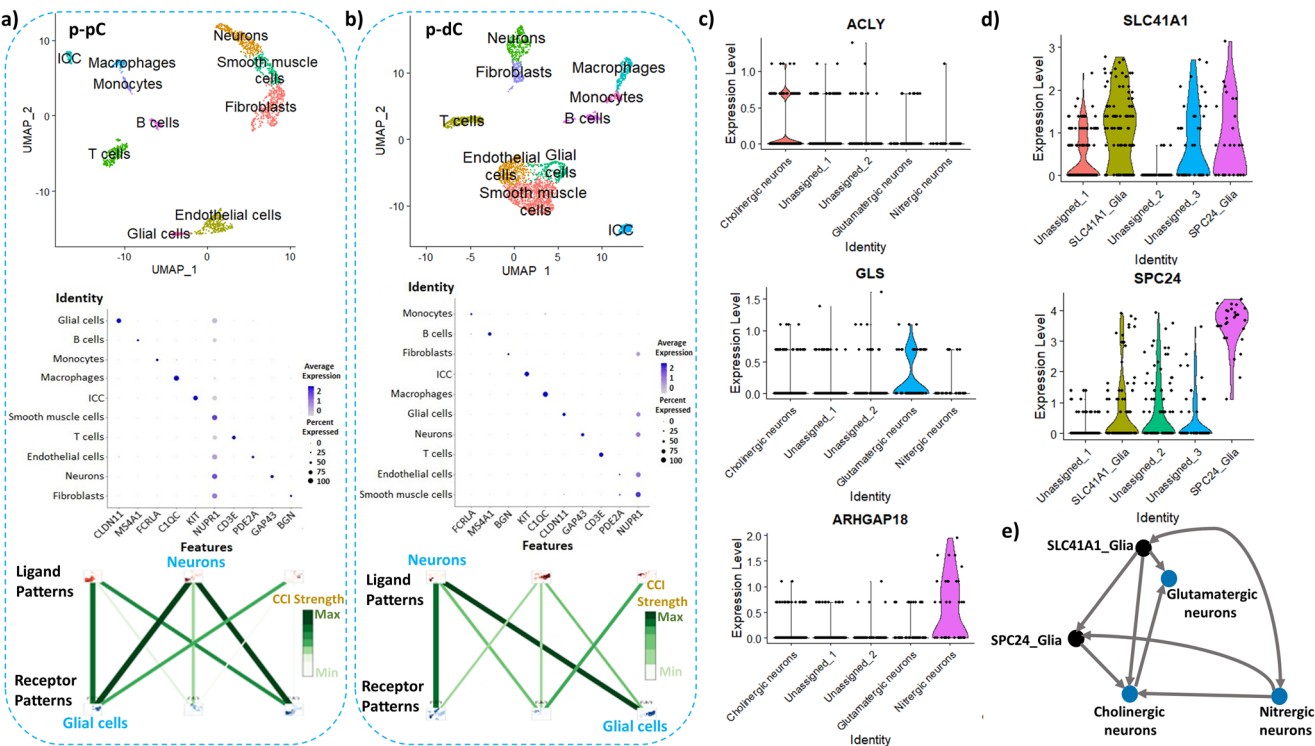

**Fig. 4 Visualization of cell types and their interaction strength in the colonic muscularis externa of naive pigs by single-cell RNA sequencing.** The clusters were labeled according to their markers and only neuronal and glial cell markers were verified by in situ hybridization in the pig proximal colon (p-pC) **a** and pig distal colon (p-dC) **b**. Uniform Manifold Approximation and Projection (UMAP) visualized 10 distinct cell clusters and the cell-cell interaction (CCI) strength between neurons and glial cells was highlighted only. Color represents a CCI strength distribution from maximum (green) to minimum (white). The distinct cell clusters were derived from 2091 cells of one p-pC individual and from 3918 cells of one p-dC individual. Joint visualization based on four individuals of p-pC and p-dC could be found in Supplementary Figure 3. Violin plots depict the distribution of gene expression levels for 3 neuronal subpopulations **c** and 2 glial subpopulations **d**. Interactions (arrows) from ligand- to receptor-expressing neuronal and glial subsets **e**. Blue, neuronal subset; Black, glial subset. Growth associated protein 43 (GAP43), a marker for annotating pan-neurons; Claudin 11 (CLDN11) for pan-glial cells; ATP citrate lyase (ACLY) for cholinergic neurons; Glutaminase (GLS) for glutamatergic neurons; Rho GTPase activating protein 18 (ARHGAP18) for nitrergic neurons; Solute carrier family 41 member 1 (SLC41A1) and SPC24 component of NDC80 kinetochore complex (SPC24) for two subtypes of glia.

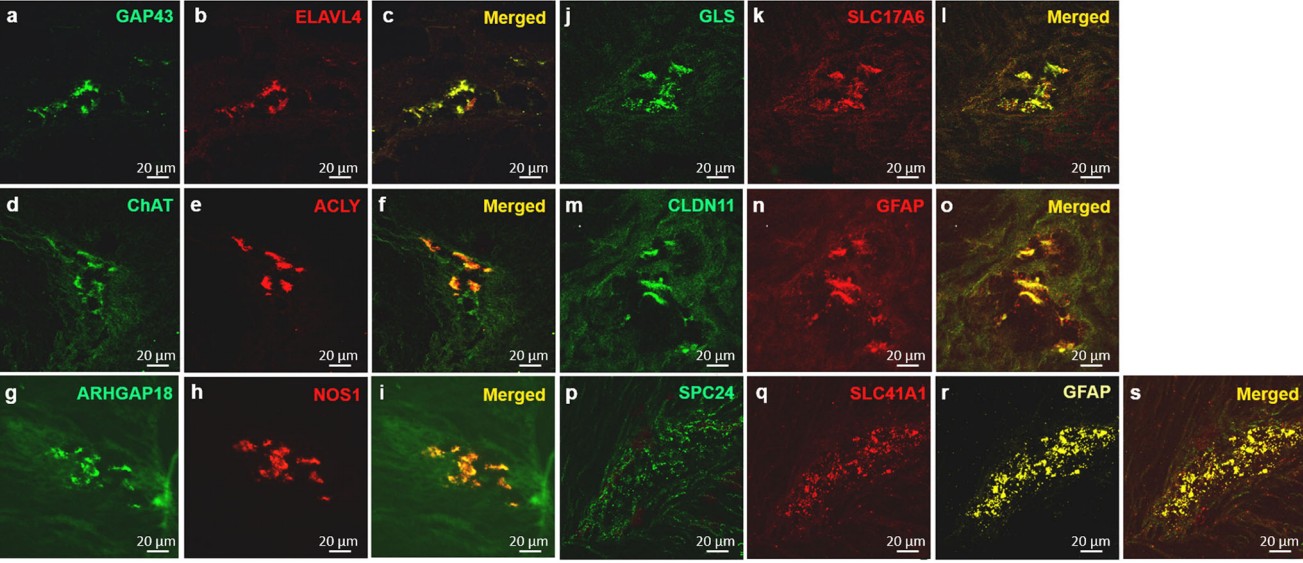

**Fig. 5 Validation of discriminatory marker genes for major classes of enteric neurons and two subtypes of enteric glial cells (EGCs) in the myenteric ganglia of porcine proximal colon using RNAscope in situ hybridization.** Confocal images were generated from 5–10 optical sections (Z-stack) cross the myenteric ganglia with frame 436 × 436 μm and 1 μm apart (20× objective). The co-expression of mRNA for *GAP43* (annotating enteric neurons) with *ELAVL4* encoding Hu C/D **a**, **b**, *ACLY* (annotating cholinergic neurons) with *ChAT* **d**, **e**, *ARHGAP18* (annotating nitrergic neurons) with *NOS1* **g**, **h**, *GLS* (annotating glutamatergic neurons) with *SLC17A6* encoding VGLU2 **j**, **k**, *CLDN11* (annotating EGCs) with *GFAP* **m**, **n**, and *SPC24*, *SLC41A1* (annotating two subtypes of EGCs) with *GFAP* **p**, **q**, **r** were indicated in the merged images **c**, **f**, **i**, **l**, **o** and **s**, respectively. Scale bars: 20 μm.

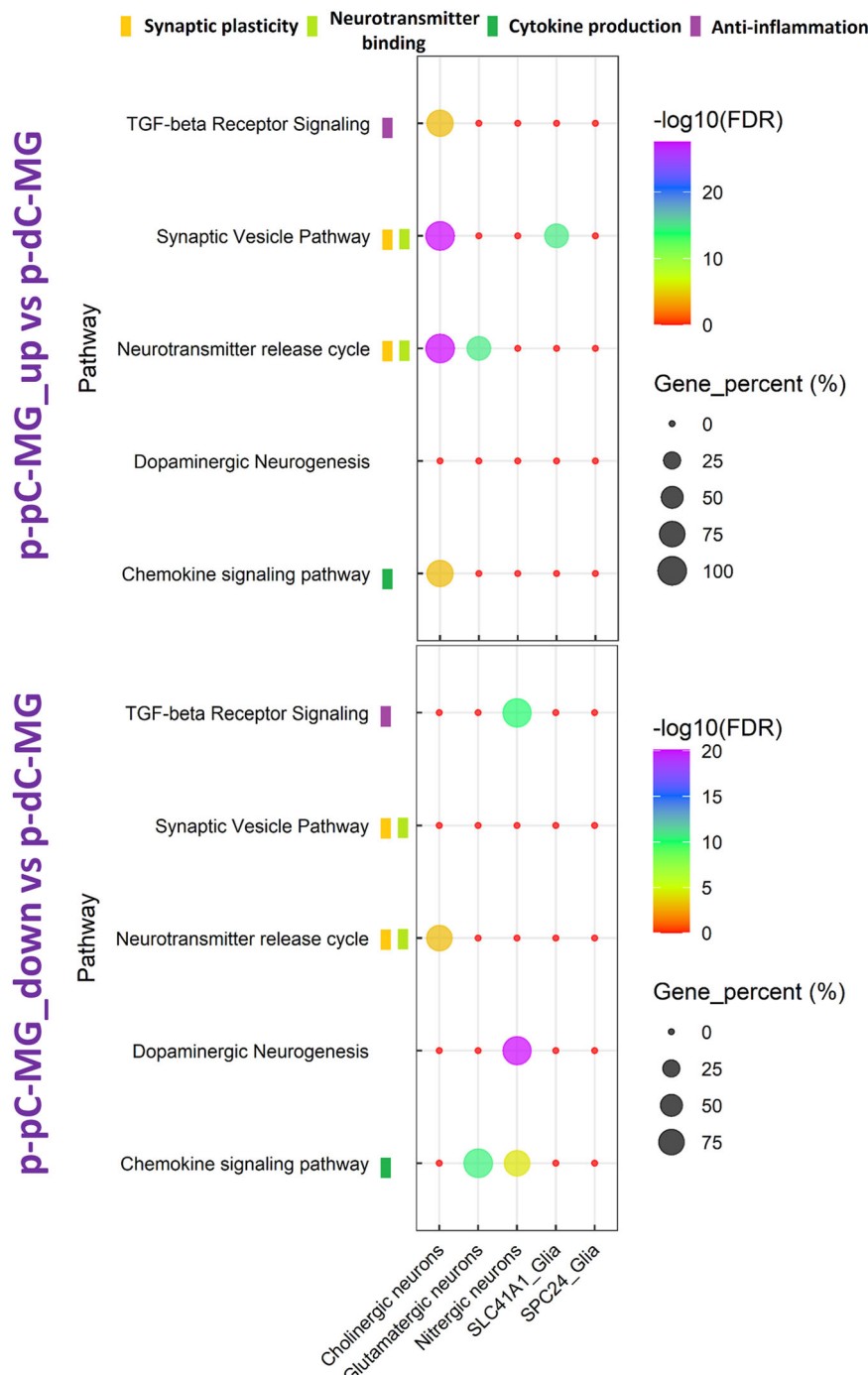

**Fig. 6 Comparison of cell-type specific pathway enrichment in myenteric ganglia (MG) between porcine proximal and distal colon (p-pC, p-dC). The bubble plot shows the gene percentage and enrichment of WikiPathways.** Circle size represents the ratio of the number of cell-type pathway-specific differentially expressed genes (DEG) and the number of total DEG in each DEG list (Gene_percent). Color represents a -log10 (FDR) distribution from big (orange) to small (purple). The color bars mean the specified WikiPathway classified into designated clusters. Up, upregulated expression. Down, downregulated expression.

of the BPs involving upregulated genes in p-pC-MG than those involving downregulated genes in p-pC-MG. However, there were some alterations of BP enrichment patterns with VNS based on comparison of p-pC-MG and p-dC-MG. Enrichment scores of the BPs such as G protein-coupled receptor signaling pathway, anatomical structure morphogenesis and system process, involving upregulated genes in p-pC-MG, were higher than those involving downregulated genes in p-pC-MG (Supplementary Figs 1, 2, 4). These BPs related to regulation of signaling, cell

communication, immune response and immune system process regulated by VNS.

Based on comparison of p-pC-MG and p-dC-MG, VNS promoted enrichment of most of the WikiPathways in p-pC-MG, including gap junctions, pro- and anti-inflammatory signaling, chemokine signaling pathway, neurotransmitter uptake and metabolism in glial cells, NO/cGMP/PKG mediated neuroprotection, dopaminergic neurogenesis, nicotine activity on dopaminergic neurons, oncostatin M signaling pathway and

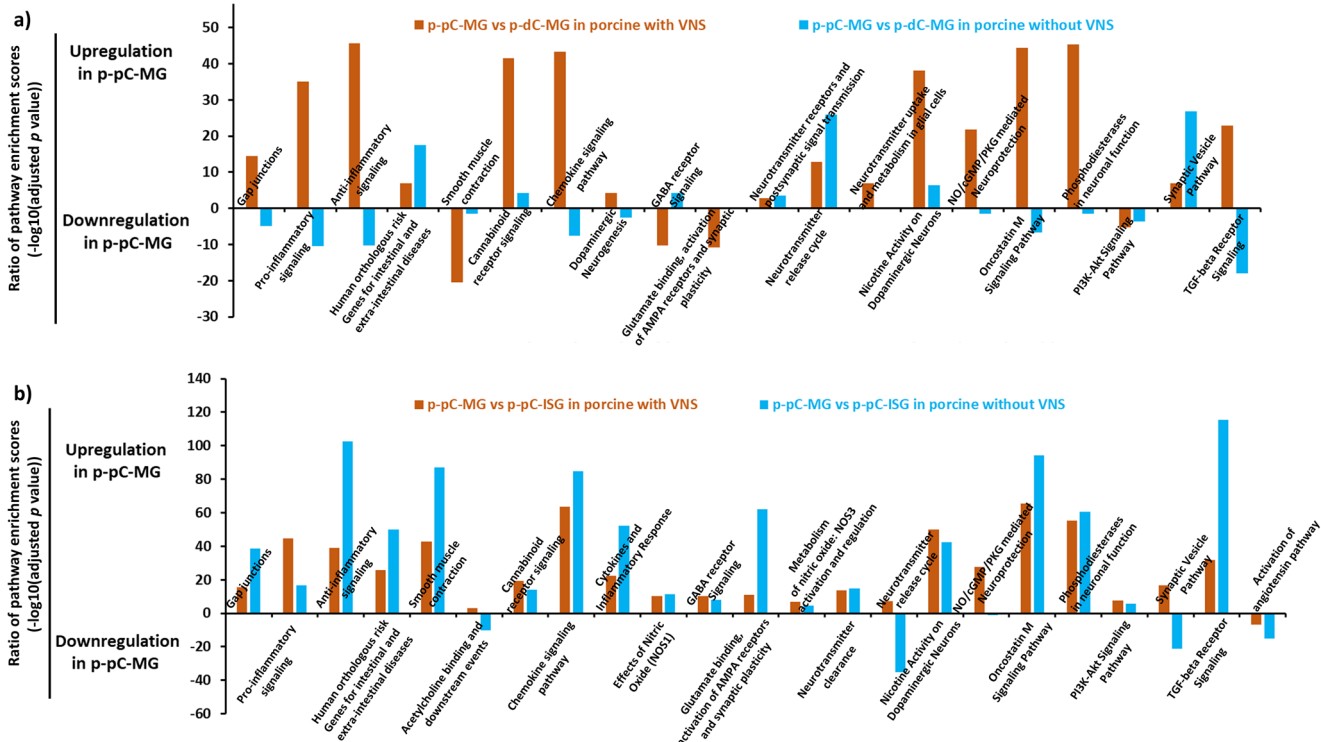

**Fig. 7 Comparison of pathway enrichment in myenteric ganglia (MG) between porcine proximal and distal colon (p-pC, p-dC), and between MG and inner submucosal ganglia (ISG) with and without vagal nerve stimulation (VNS).** The figures show the enrichment ratio of the same WikiPathway involving upregulated and downregulated genes in each list of differentially expressed genes based on the bulk RNA sequencing data from MG of p-pC and p-dC **a** and MG and ISG of p-pC **b**. The light blue color represents the enrichment ratio without VNS and the orange color represents the enrichment ratio with VNS.

TGF-beta receptor signaling, and reduced enrichment of human orthologous risk genes for intestinal and extra-intestinal diseases, synaptic vesicle pathway and neurotransmitter release cycle (Fig. 7a, Supplementary Data 2). By contrast, effects of VNS on p-dC-MG were reflected by improved enrichment of smooth muscle contraction, GABA receptor signaling and glutamate binding, activation of AMPA receptors and synaptic plasticity (Fig. 7a). In addition, the comparison between p-pC-MG and p-pC-ISG showed that VNS decreased the enrichment scores of most of the WikiPathways in p-pC-MG, such as gap junctions, anti-inflammatory signaling, smooth muscle contraction, chemokine signaling pathway, glutamate binding, activation of AMPA receptors and synaptic plasticity and TGF-beta receptor signaling, while increasing the enrichment scores of the WikiPathways, including pro-inflammatory signaling, acetylcholine binding and downstream events, GABA receptor signaling, neurotransmitter release cycle and synaptic vesicle pathway (Fig. 7b). The alterations were further explained by the functional similarity between gene products, which was measured by semantic similarities between the annotated BPs of each gene, resulting in similarity matrix (Supplementary Fig. 6). High average similarity scores in the WikiPathways of interest indicate strong functional similarities. Gap junctions and neurotransmitter release cycle had the highest similarity scores. High scores existed when comparing acetylcholine binding and downstream events or neurotransmitter release cycle with pro-inflammatory unlike anti-inflammatory signaling. Apart from functional similarities, the BPs comprised in the WikiPathways were interactive. Pearson correlation analysis revealed a significant positive correlation among gap junctions, smooth muscle contraction and anti-inflammatory signaling and between acetylcholine binding and downstream events and neurotransmitter release cycle. Finally, VNS exerted its effects on p-pC-MG by reducing the enrichment

of human orthologous risk genes for intestinal and extra-intestinal diseases and on p-pC-ISG by decreasing the activation of angiotensin pathway, respectively (Fig. 7b, Supplementary Data 2).

Cell-type contribution to the enrichment of the WikiPathways under VNS was further evaluated based on the comparison of p-pC-MG and p-dC-MG (Fig. 8, Supplementary Data 3). The functional linkages involving all cell-type specific DEG that mediated the top five WikiPathways covered >90% of those involving all DEG in each comparison with or without VNS (Supplementary Table 2). In cholinergic neurons of p-pC-MG, VNS substantially increased the enrichment of all top five WikiPathways including chemokine signaling pathway, oncostatin M signaling pathway, TGF-beta receptor signaling, gap junctions and NO/cGMP/PKG mediated neuroprotection. By contrast, VNS reduced the enrichment of gap junctions in nitrergic neurons and SLC41A1_Glia, while increasing those of chemokine signaling pathway in SPC24_Glia. In addition, VNS increased the enrichment scores of oncostatin M signaling pathway in both glutamatergic and nitrergic neurons in p-pC-MG. VNS exerted its effects on glutamatergic neurons in p-dC-MG by increasing enrichment of TGF-beta receptor signaling and decreasing enrichment of chemokine signaling pathway. The enrichment ratio of the shared BPs between the investigated WikiPathways and pro- or anti-inflammatory signaling was calculated to assess the contribution of the pathway enrichment alterations to inflammatory status under VNS. Under VNS, cholinergic and glutamatergic neurons and SLC41A1_Glia contributed to anti-inflammation, while only nitrergic neurons contributed to inflammation in p-pC-MG. By contrast, anti-inflammation was attributed to glutamatergic and nitrergic neurons and SLC41A1_Glia in p-dC-MG (Fig. 9).

The response of cell-cell interactions to VNS was estimated through differential expression of genes encoding ligands and

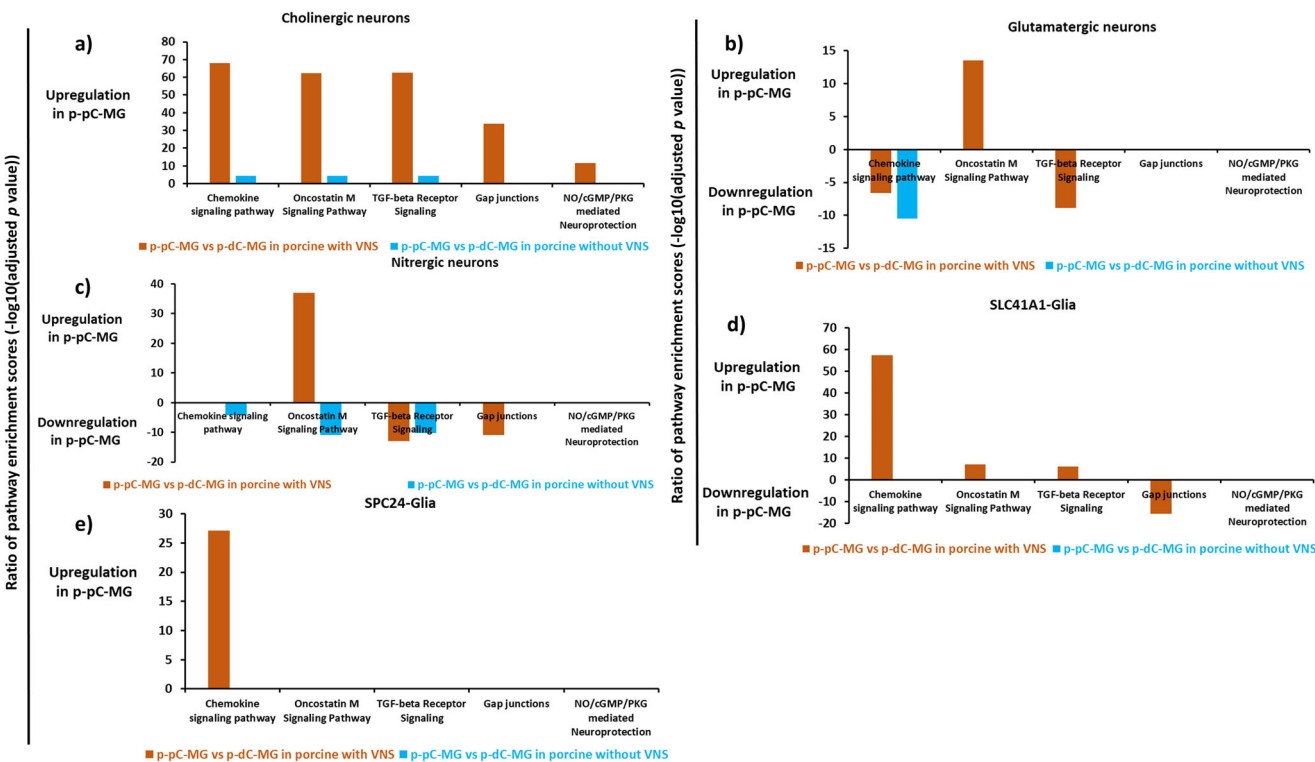

**Fig. 8 Comparison of cell-type pathway enrichment in myenteric ganglia (MG) of porcine proximal and distal colon (p-pC, p-dC) with and without vagal nerve stimulation (VNS).** The figures show the enrichment ratio of the same WikiPathway involving upregulated and downregulated genes in each list of differentially expressed genes and only show the enrichment ratio of the top five WikiPathways in cholinergic neurons **a**, glutamatergic neurons **b**, nitrergic neurons **c**, SLC41A1-glia **d**, and SPC24- glia **e**. The light blue color represents the cell-type enrichment ratio without VNS and the orange color represents the cell-type enrichment ratio with VNS.

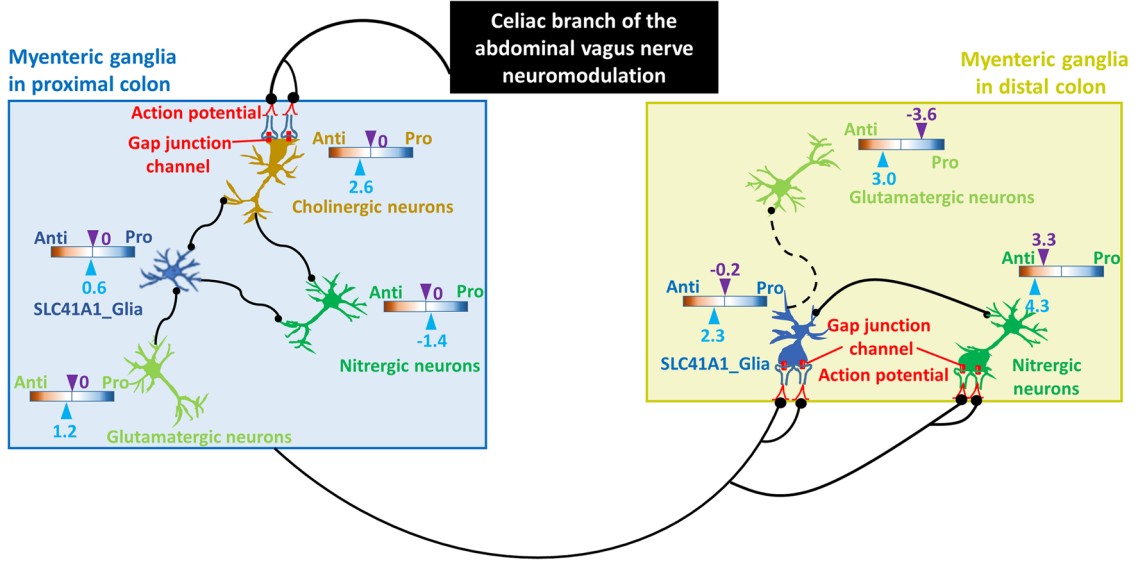

**Fig. 9 Cartoon diagram illustrating potential electrical transmission along the proximal-distal axis of the porcine colon under vagal nerve stimulation (VNS) and the response mechanisms of certain neuronal/glial cell populations to VNS.** The enrichment ratio of the shared biological processes (BPs) between the top five WikiPathways and pro- or anti-inflammatory signaling (pro/anti) was calculated. The solid and dot lines represent cell–cell interactions with and without alterations under VNS, respectively. Black dots represent the targeting sites. Color represents a log10(pro/anti) distribution from Anti (orange) to Pro (blue). Arrowheads in light blue and purple indicate the enrichment ratio with and without VNS, respectively. *Pro* pro-inflammation, *Anti* anti-inflammation.

**a)**

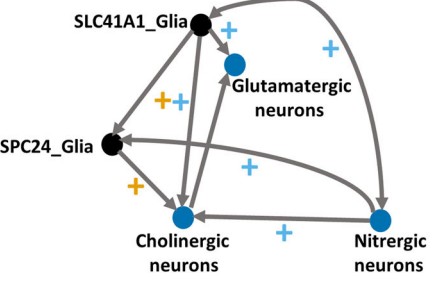

**p-pC-MG up vs p-dC-MG in pig with VNS**

| Gene | LR | Cell type | p-value |
|---|---|---|---|
| LDLR | Receptor | Cholinergic neurons | 2.17E-05 |
| VEGFA | Ligand | Nitrergic neurons | 4.08E-05 |
| TGFB3 | Ligand | SLC41A1_Glia | 6.03E-07 |

**p-pC-MG up vs p-dC-MG in pig without VNS**

| Gene | LR | Cell type | p-value |
|---|---|---|---|
| LDLR | Receptor | Cholinergic neurons | 0.00117 |
| VEGFA | Ligand | Nitrergic neurons | > 0.05 |
| TGFB3 | Ligand | SLC41A1_Glia | > 0.05 |

**b)**

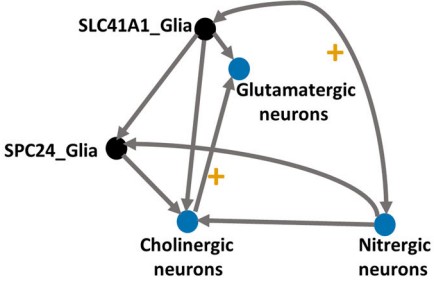

**p-pC-MG down vs p-dC-MG in pig with VNS**

| Gene | LR | Cell type | p-value |
|---|---|---|---|
| GPC4 | Receptor | Glutamatergic neurons | 0.0115 |
| GNG8 | Receptor | Nitrergic neurons | 1.98E-05 |
| CALCRL | Receptor | SLC41A1_Glia | 0.0311 |

**p-pC-MG down vs p-dC-MG in pig without VNS**

| Gene | LR | Cell type | p-value |
|---|---|---|---|
| GPC4 | Receptor | Glutamatergic neurons | > 0.05 |
| GNG8 | Receptor | Nitrergic neurons | 5.00E-06 |
| CALCRL | Receptor | SLC41A1_Glia | > 0.05 |

**Fig. 10 The response of cell-cell interactions to vagal nerve stimulation in neuronal and glial subsets.** The alterations were estimated through differential expression (p-value) of ligands (sky blue) and receptors (orange) expressed in neuronal and glial subsets in myenteric ganglia (MG) with vagal nerve stimulation (VNS) compared with those without VNS. + labels the upregulated interaction of cell subsets where there are the genes expressing ligands/receptors with smaller p-value under VNS. The upregulated **a** and downregulated **b** genes encoding both ligands and receptors in myenteric ganglia (MG) with or without VNS were compared between the porcine proximal and distal colon (p-pC, p-dC).

receptors in neuronal and glial subsets (Figs. 9 and 10). VNS potentially improved the cell-cell interactions in p-pC-MG, including the interactions between SLC41A1_glia and cholinergic or glutamatergic or nitrergic neurons, cholinergic neurons and nitrergic neurons, and SPC24_glia and cholinergic or nitrergic neurons. The interactions between SLC41A1_glia and cholinergic neurons may be stronger because of the upregulated expression of the genes encoding both ligands and receptors. VNS impacted on p-dC-MG by potentially improving the interactions between cholinergic and glutamatergic neurons and between SLC41A1_glia and nitrergic neurons (Figs. 9 and 10).

## Discussion

Even though the porcine gut is increasingly regarded as a useful translational model, the knowledge of the molecular profiling of porcine ENS and its similarity to that of human is still lacking. This study demonstrated the distinct transcriptional programs associated with functional characteristics existing in the porcine colonic enteric ganglia along different regions using bulk and single-cell RNA-seq. The cross-comparison of the regional transcriptomic profiling revealed a high conservation of these transcriptional programs associated with functional characteristics in the colonic myenteric ganglia between pig and human. Interestingly, we found that these programs explained more than 96% of their transcriptomic responses to VNS. These findings suggest that porcine colon could serve as predictors in translational studies.

Recent studies have identified several conserved marker genes in ENS cell subtypes between human and mouse based on orthologous gene expression[18,28]. According to our bulk RNA-seq data from both porcine and human ENS, we uncovered 7246 conserved genes between h-aC-MG and p-pC-MG (p-value > 0.001) and 2601 between of h-dC-MG and p-dC-MG (p-value > 0.001) from a total of 12,291 high-quality orthologous genes using R package SCBN. The functional linkages of gene regulatory networks, where single genes are positioned, have been used as powerful indicator, reflecting mechanistic insights and functional similarities across species[29–32]. Pathway analysis could identify biological pathways that are enriched in DEG lists and highlight the overlaps among pathways based on the shared genes[29]. We extracted the gene networks to indicate the differentiation between h-aC-MG and h-dC-MG and between p-pC-MG and p-dC-MG. Twelve porcine orthologous genes from comparison of h-aC-MG and h-dC-MG were found, of which only 2 genes were the conserved genes based on the outputs from R package SCBN. Nonetheless, the 12 orthologous genes served as driver genes, whose networks represented all the characteristics of p-pC-MG and p-dC-MG in the pig, in terms of 100% linkage coverage (Fig. 2). It was worth noting that the networks derived from the 12 porcine orthologous driver genes illuminated >96% functional alterations in p-pC-MG and p-dC-MG of porcine subjected to VNS (Fig. 2d). The networks mediated by the human orthologous genes from comparison of p-pC-MG and p-dC-MG partially overlapped those derived from the DEG lists in comparison of h-aC-MG and h-dC-MG, probably because of the lack of complete annotation and assembly of the porcine genome[33]. Thus, although there were few conserved genes between porcine and human colon, the networks of interacting orthologous genes orchestrated almost the same complex colonic functions across species, and p-pC and p-dC could serve as transcriptional predictors, with important implications in understanding of human colonic functions and neuromodulation.

Our work also shows the ENS transcriptomic heterogeneity between MG and ISG along the proximal-distal axis of the porcine colon. Comparisons of MG and ISG in each of p-pC and p-dC uncovered several pathways responsible for normal colonic physiological functions such as peristalsis and secretion. The inflammatory signaling pathways prevailed in MG, including interactions between immune cells and neurons (e.g., oncostatin M signaling pathway and cytokines and inflammatory response), cytokine production (e.g., chemokine signaling pathway) and anti-inflammation (e.g., TGF-beta receptor signaling) (Fig. 3a, b).

Of them, the enrichment of anti-inflammation, involving >99% of the DEG, was the highest. The reason for the inflammatory signaling pathways predominant in MG is probably because the inflammatory bowel disease (IBD) risk genes (e.g., *SMAD3* and *OSMR*) were significantly expressed in MG. Our data also showed that all inflammatory signaling pathways closely interacted with smooth muscle contraction, indicating that the inflammatory status influenced smooth muscle activity. In addition, effects of nitric oxide mediated by *nNOS* and neurotransmitter clearance mediated by *SLC6A4* were dominant in MG. Both nNOS and SLC6A4 that lowers the levels of serotonin are responsible for relaxation of smooth muscle[34–36]. Furthermore, the enrichment of glutamate binding, the activation of AMPA receptors and synaptic plasticity was much higher in MG than in ISG and directly interacted with the BPs such as the regulation of blood circulation and smooth muscle contraction in MG. By contrast, ISG had higher enrichment of activation of angiotensin pathway, synaptic vesicle pathway, acetylcholine synthesis, binding and downstream events and dopamine transport and secretion (Fig. 3a, b), all of which interacted with each other. Angiotensinogen (AGT) that was located in the core of activation of angiotensin pathway and selected in our study has been involved in the development of GI mucosal injury due to SARS-CoV-2 and other factors[37,38]. Therefore, we used the enrichment of this pathway to evaluate the mucosal integrity. The results showed that only ISG contributed to the modulation of mucosal integrity. Moreover, we found that the activation of angiotensin pathway, acetylcholine synthesis, binding and downstream events and dopamine transport and secretion shared the BPs namely signaling and cell communication, suggesting the potential roles of acetylcholine-dopamine balance in maintenance of mucosal integrity. It is worth noting that neurotransmitter uptake and metabolism in glial cells, mediated by SLC1A3, a glial high affinity glutamate transporter, was dominant in MG only in p-dC (Fig. 3a, b). A steep oxygen gradient along the proximal-distal axis of colon exists, which leads to formation of reactive oxygen species and the resulting glutamate excitotoxicity[39–41]. Glutamate uptake in glia guarantees the neural circuit integrity because SLC1A3 connected the BPs such as synaptic signaling and neurogenesis.

When comparing MG between p-pC and p-dC, we found that the inflammatory signaling pathways such as cytokines and inflammatory response, chemokine signaling pathway and TGF-beta receptor signaling were prevalent in p-dC-MG (Fig. 3c). There is evidence that hypoxia and inflammation are intertwined[41]. In addition, p-dC-MG compared to p-pC-MG had much higher enrichment of smooth muscle contraction, NO/cGMP/PKG mediated neuroprotection and dopaminergic neurogenesis (Fig. 3c), with all of them having interactions. It has been reported that GUCY1A1 (guanylate cyclases), which was involved in NO/cGMP/PKG mediated neuroprotection, is activated by nitric oxide and plays a key role in inhibiting neuroinflammation[42]. From this perspective, we infer that the combination of nitrergic and dopaminergic neurons contributes to the regulation of smooth muscle contraction. This was confirmed by our scRNA-seq data showing that the anti-inflammatory status, determined by more BPs shared by TGF-beta signaling, chemokine signaling pathway and anti-inflammatory signaling, was closely related to the differentiation of two neuronal populations (Fig. 6). This is consistent with recent reports, implicating the role of TGF-beta in differentiation of nitrergic and catecholaminergic enteric neurons[43,44]. Recent study showed that the interactions of these two neuronal populations could stabilize asynchronous contractile activity and lead to the generation of colonic peristalsis in proximal colon[45]. However, our results suggested that the contribution of

dopaminergic and nitrergic neurons to colonic peristalsis is much greater in p-dC than in p-pC. By contrast, p-pC-MG had more complex neuronal wiring, highlighted by the high enrichment of nicotine activity on dopaminergic neurons (mediated by *TH*), acetylcholine synthesis (mediated by *ChAT*), synaptic vesicle pathway, neurotransmitter receptors and postsynaptic signal transmission (mediated by *HTR3A*), neurotransmitter release cycle and GABA receptor signaling (mediated by *GABRB3* and *GABRA4*) (Fig. 3c). Cholinergic neurons played leading roles based on their contribution to the top five WikiPathways (Fig. 6), which is in accordance with the findings of Li et al[15].

The comparison of ISG between p-pC and p-dC revealed that the inflammatory signaling pathways such as chemokine signaling pathway and TGF-beta receptor signaling prevailed in p-dC-ISG, while in p-pC-ISG, there was higher synaptic activity, characterized by the high enrichment of nicotine activity on dopaminergic neurons (mediated by *TH*) and neurotransmitter receptors and postsynaptic signal transmission (mediated by *HTR3A* and *HTR3B*) (Fig. 3d). However, we also found high enrichment of blood vessel development and effects of nitric oxide (mediated by *nNOS*) in p-dC-ISG. Nerve fibers may contribute to the perivascular circulation based on the distribution of nNOS and its catalytic activity with nNOS-derived NO maintaining vasodilation[46]. Unlike the comparison of MG between p-pC and p-dC, dopaminergic neurogenesis and dopamine biosynthetic process were predominant in p-pC-ISG. Evidence showed that dopamine could promote colonic mucus secretion[47].

Our previous study has indicated that the functional sphere of the influence of VNS on porcine colon was pan colonic, from p-pC to p-dC, and VNS increased contractions across the colon[17]. The evidence also showed that the vagal nerve endings often synapse onto neurons in the myenteric plexus[48]. Consistent with this report, our data showed that the major changes induced by VNS happened in MG, especially in p-pC-MG. Although VNS led to some differences of pathway enrichment patterns when comparing p-pC-MG with p-pC-ISG, the alterations were not as prominent as those in comparison of p-pC-MG and p-dC-MG characterized by the enrichment pattern reversal of more than half the investigated pathways (Fig. 7). Compared with p-pC-MG, the obvious changes in p-dC-MG were the enhanced enrichment of smooth muscle contraction and glutamate binding, activation of AMPA receptors and synaptic plasticity, which were interactive. This suggested the crucial roles of synaptic plasticity mediated by glutamate in regulation of smooth muscle contraction. Interestingly, cholinergic neurons contributed to the enrichment of glutamate binding, activation of AMPA receptors and synaptic plasticity, implying that cholinergic neurons may co-release both acetylcholine and glutamate, and the released glutamate could activate postsynaptic neurons (glutamatergic neurons)[49]. Our cell–cell interaction data also confirmed the potentially upregulated connection between cholinergic and glutamatergic neurons under VNS (Fig. 10b). Together with the improved enrichment of GABA receptor signaling, we speculated that VNS promoted the occurrence of rhythmic phasic contractions and fecal pellet expulsion in p-dC. However, in p-pC-ISG, there was the decreased enrichment of acetylcholine binding and downstream events, neurotransmitter release cycle and synaptic vesicle pathway. Of them, VNS led to the highest alterations in the enrichment of glutamate release cycle (*GLS/GLS2*). This may be related to the VNS-induced the activation of an anti-inflammatory response[40,41]. This is in accord with the improvement of the mucosal integrity, characterized by the reduced enrichment of activation of angiotensin pathway (Fig. 7b).

Gap junctions greatly contributes to electrical transmission[50]. Here, we used the enrichment changes of gap junctions to interpret the influences of VNS on the transcriptomic profiling.

When comparing p-pC-MG with p-pC-ISG, the enrichment of gap junctions, anti-inflammatory signaling and smooth muscle contraction was surprisingly reduced in p-pC-MG by VNS (Fig. 7b) and had a significant positive correlation (Supplementary Fig 6). In addition, we found that glutamate release cycle and gap junctions showed the highest similarity score, suggesting that glutamate transmission may complement the function of gap junctions. Consequently, glutamate release triggered acetylcholine binding and downstream events (CHRNA5) due to their significantly positive correlation (Supplementary Fig 6). The potentially synergistic functions of glutamate and acetylcholine may be important for functional modulation in p-pC-MG. Collectively, the alterations of the pathway enrichment under VNS may reduce disease risks in view of the decreased enrichment of human orthologous risk genes for intestinal and extra-intestinal diseases, associated with gut dysmotility and inflammation in p-pC-MG (Fig. 7b).

Compared with p-dC-MG, enrichment of gap junctions was highly increased in p-pC-MG, while enrichment of synaptic vesicle pathway, neurotransmitter receptors and postsynaptic signal transmission and neurotransmitter release cycle was decreased. We speculated that there is direct mechanistic relationship between the formations of neuronal gap junction coupling and the disappearance of chemical transmission in most neuronal cell types as reported[50]. Previous studies showed that VNS reduces intestinal inflammation through the cholinergic anti-inflammatory pathway[51]. In line with the statement, we found that the enrichment ratio of pro- to anti-inflammatory signaling (pro/anti) without VNS was 1.4 times higher than that with VNS, suggesting the anti-inflammatory effects of VNS (Fig. 7a). The enrichment of pro-inflammatory signaling under VNS was attributed to some extent to nitrergic neurons as supported by a recent report showing that NOS1 led to an outburst of inflammatory reactions in macrophages via activator protein-1[52]. Whereas enrichment of anti-inflammatory signaling was attributed to both cholinergic and glutamatergic neurons and SLC41A1_Glia in p-pC-MG (Figs. 8 and 9). The enteric glia were thought to be essential to reduce the intestinal inflammatory response under VNS[53]. In the vagal circuitry, glutamate is a primary neurotransmitter involved in key gastrointestinal functions[54]. However, its anti-inflammatory effects are rarely reported except that excessive release of glutamate into the extrasynaptic space led to inflammatory responses, which could be corrected by glial cells in central nervous system[55]. Our result indicated that the synergies of glutamatergic neurons with enteric glia were also potentially present in ENS under VNS because of the increased enrichment of neurotransmitter uptake and metabolism in glial cells (GLUL, conversion of glutamate and ammonia to glutamine) in p-pC-MG (Fig. 7a). Duo to the direct interactions between smooth muscle contraction and anti-inflammatory signaling, the anti-inflammatory properties indeed reduced the enrichment of human orthologous risk genes for intestinal and extra-intestinal diseases, associated with gut dysmotility and inflammation (Fig. 7a). We found that OSMR, the only risk gene for IBD was expressed significantly in p-pC-MG compared to p-dC-MG under VNS. OSMR was enriched in SLC41A1-Glia and the OSMR-mediated pathways shared BPs with pro- and anti-inflammatory signaling. The enrichment ratio of the shared BPs (pro/anti) was 1.4 with VNS, while pro/anti was 1.2 without VNS, which did not show the significant anti-inflammatory effects of VNS on IBD. Future work should involve more risk genes for IBD to convincingly identify the exact correlation between vagal function and IBD. Finally, we derived the VNS influences on p-dC-MG from the improved enrichment of gap junctions in SLC41A1-Glia and nitrergic neurons of p-dC-MG, compared with p-pC-MG (Figs. 8d, 9). Surprisingly, unlike

the pro-inflammatory impacts of nitrergic neurons in p-pC-MG, they played the anti-inflammatory roles in p-dC-MG, together with glutamatergic neurons and SLC41A1_Glia. Moreover, the contribution of SLC41A1_Glia to anti-inflammation was much higher in p-dC-MG. Interestingly, the effects of VNS on glutamatergic neurons and SLC41A1_Glia in p-dC-MG led to switch of their contribution from inflammation to anti-inflammation (Fig. 9). Different from the attribution of glutamatergic neurons to anti-inflammation in p-pC-MG under VNS, the increased enrichment of glutamate binding, activation of AMPA receptors and synaptic plasticity in cholinergic neurons ensured the correction of glutamatergic neurotransmission dysfunction in p-dC-MG (Fig. 7a). Thus, we inferred that the synergies of glutamatergic neurons with cholinergic neurons contributed to anti-inflammation in p-dC-MG.

Some limitations in this study should be noted. We prepared 9 cell suspensions of the muscularis externa from each p-pC, p-tC and p-dC of 3 naive adult pigs which yielded an average of 1930 (735/2091/2965), 2318 (1993/1903/3057) and 2559 (1381/3918/2378) mapped single cells in each of p-pC, p-tC and p-dC respectively and generated 9 scRNA-seq datasets with 10 cell clusters including neurons and glia. The total mapped cells for scRNA-seq analysis are less than those reported in the mouse and human[18,28]. According to the R toolkit Seurat for scRNA-seq data analysis (Satija Lab: https://satijalab.org/howmanycells/), at least 702 cells should be mapped if ten cell types needed to be clustered and each one was present at a fraction of 2% of the total population under the condition of 95% confidence that the sample contained at least 6 cells from each of those cell types. Thus, the numbers of mapped cells for scRNA-seq analysis in present study are above the minimum recommended by Sataija Lab. In addition, it has been proved that as few as 40,000 reads per cell should be enough for unbiased cell-type classification within a mixed population of distinct cell types[56,57]. Furthermore, Zhang et al.[58] identified ~1560 and ~6478 cells through two runs of 10× Genomics. The results indicated that the distribution curves of cells' nearest correlation with all other cells from the two runs overlapped, suggesting that two different cell inputs achieved the similar cell clusters. The nearest correlation is the basis of many clustering or classification strategies and the R toolkit Seurat also uses this strategy. To sum up, the total mapped single cells in present study are still much enough for us to draw the solid conclusions. Among the mapped cells, three major neuronal (cholinergic, nitrergic, and glutamatergic) and two glial subpopulations were annotated with discriminatory marker genes and validated using RNAscope in situ hybridization. The subpopulations and sample sizes of neurons and glia are relatively small in our scRNA-seq study, but the cell number in each subpopulation was still greater than 127, the minimum recommended by Satija Lab on the basis of the criteria that five cell types needed to be clustered and each one was present at a fraction of 10% of the total population under the condition of 95% confidence that the sample contained at least 6 cells from each of those cell types. The low number of neurons and glia is largely due to the challenge to isolate these cells, in particular the neurons in the MG from adult pig colon. Myenteric neurons are sparse, dispersed among other cell types (e.g., myocytes and fibroblasts) and are sandwiched between two muscle layers within the muscularis externa of the colon wall which limited our ability to isolating intact myenteric neurons and enrich populations in single cells. Recent studies of the ENS using scRNAseq have also been restricted to a limited number of myenteric neurons despite these myenteric neurons were isolated from embryonic animals or the early post-natal mice and human[18,28] rather than adult animals. There is a clear need for us to optimize the digestion and disaggregation of adult pig colonic tissue samples to consistently

obtain high quality, viable, single myenteric neurons and glia. In addition, the validation for putative cell–cell interactions requires extensive in situ hybridization. The effects of VNS on transcriptional responses in different regions of human colon also need to be confirmed in clinical studies and clarification of sex differences should involve the intact animals and large human group sizes.

In summary, this study compared the transcriptomic profiles in the colonic ENS between pig and human and across two layers of enteric plexus along the proximal to distal colonic regions in pigs. The regional-specific gene programs were identified at both bulk and single-cell resolution in pigs and regional-dependent highly conserved core transcriptional programs were revealed between the pig and human. VNS exerted its effects mainly on the myenteric ganglia in pigs. Regional-specific transcriptomic responses to VNS shared >96% of the conserved core transcriptional programs between the pig and human. These findings provide a fundamental resource for better understanding of the importance of porcine model in the colonic translational research.

## Methods

**Tissue collections**. Six castrated male and three intact female Yucatan minipigs (~7 months old, 25–36 kg, S&S Farms, Ramona, CA) were fasted for 12 h and anaesthetized by intramuscular application of midazolam (1 mg/kg, cat # 067595, Covetrus, Dublin, OH), ketamine (15 mg/kg, cat # 068317, Covetrus, Dublin, OH) and meloxicam (0.3 mg/kg, #049755, Covetrus, Dublin, OH). Three out of the six male animals underwent electrical stimulation of the celiac branch of the abdominal vagus nerve (2 Hz, 0.3–4 ms, 5 mA, 10 min) using pulse train. A detailed experimental protocol for VNS is available at https://www.protocols.io/view/tache-mulugeta-ot2od024899-colon-tissue-electrical-3rmgm. The colonic specimens with full thickness (about 1.5 cm long) were removed from the proximal colon (p-pC, about 10 cm from the ceco-colic junction), transverse colon (p-tC, about 10 cm from the end of the proximal, specifically about 10 cm from the end of the centrifugal spiral colon) and distal colon (p-dC, about 20 cm from the ano-rectum). Time intervals between anesthesia injection and tissue collections were 20–30 min for naive pigs without VNS and about 5 h for those with VNS. All samples were embedded in OCT, snap-frozen in dry ice and stored at −80 °C for LCM procedures. One extra piece (4 × 4 cm) of full thickness colon tissue samples was harvested from each p-pC, p-tC and p-dC of 3 naive pigs (2 males and 1 female) and processed for scRNA-seq. All animal care and procedures were performed following National Institutes of Health guidelines for the humane use of animals, with approval and in accordance with the guidelines of the University of California at Los Angeles (UCLA) Institutional Animal Care and Use Committee, Chancellor's Animal Research Committee (ARC) (protocol 2018-074-01). All efforts were made to avoid suffering.

The full thickness of human colonic specimens (about 5 × 3 cm) were dissected from healthy margin of the post-operative ascending, transverse and descending colon (h-aC, h-tC, h-dC, 4 of each) from 12 patients (5 males and 7 females, median age 47 years old, range 35–66 years) with colonic adenocarcinoma. All specimens were provided by the UCLA Translational Pathology Core Laboratory after dissection and examination to be normal under macro- and microscopic inspection. None of the patients had active colonic infections when the tissues were collected. Samples were immersed in Belzer UW® Cold Storage Solution (Bridge to Life Ltd, Columbia, SC) on ice. Once delivery, samples were instantly embedded in OCT, snap-frozen in dry ice and stored at −80 °C for LCM procedures. The interval time between colonic resection and snap freezing was 70–90 min. The use of human colon tissues was approved by the UCLA Institutional Review Board for Biosafety and Ethics (IRB #17-001686). Informed consent has been obtained in all cases.

**Laser-capture microdissection (LCM) of enteric ganglia from human and porcine colon and bulk RNA-seq**. A range of 25–40 ganglia/subject were harvested from ISG and MG respectively of p-pC, p-tC and p-dC (6 without VNS and 3 with VNS) and from MG of h-aC, h-tC and h-dC (12 subjects) using LMD-6000 Laser Micro-dissection System (Leica Microsystems, Wetzlar, Germany). The details were as follows. Full thickness colon tissue samples were collected from each p-pC, p-tC and p-dC of pig and from each h-aC, h-tC and h-dC of human. After washed in the ice-cold and DEPC treated PBS, the tissue samples were embedded in OCT, snap-frozen in dry ice and stored at −80 °C until cryosectioning. Tissue sections of 10 μm were cut using the Microm HM 500 M cryostat (Micron Instruments, CA, USA) at −20 °C and mounted onto PEN membrane (2.0 μm) slides (Leica Microsystems, Wetzlar, Germany) for UV LCM (Leica LMD6000, Leica Microsystems, Wetzlar, Germany). After mounting the sections, the tissue sections were stained with 1% cresyl violet stain solution (Sigma-Aldrich, MO, USA) before undergoing dehydration through graded alcohols (Thermo Fisher Scientific, NY, USA) (75%, 95%, 100%, 100%) and xylene (Sigma-Aldrich, MO, USA) for a total of 3 min and air-dry for 5 min. Tissue sections were microdissected on a UV laser-

based Leica LMD6000 laser microdissection system using the following parameters: 20× (objective), 50 (laser power), 10 (aperture), 8 (speed), 30 (specimen balance). The cutting followed the indicated marks that outlined the desired ISG and MG. We collected 25–40 ganglia per tissue section from ISG and MG in pig and from MG in human on the 0.5 ml tube cap (USA Scientific, Inc., FL, USA) filled with lysis solution from the QIAgen RNAeasy Micro Kit (Qiagen, CA, USA). Following LCM, total RNA was extracted using the same kit according to the manufacturer's instructions. All RNA samples were stored in nuclease-free tubes at −80 °C. The quality and quantity of RNA samples were checked using Agilent RNA 6000 Pico Kit (Agilent, CA, USA) on Agilent 2100 Bioanalyzer (Agilent, CA, USA). The quality of RNA samples (67 samples) with RIN over 6 were used to construct cDNA libraries using SMART-Seq® Stranded Kit (Takara Bio USA, Inc., CA, USA), whose quality and quantity were checked using Agilent High Sensitivity DNA Kit on Agilent 2100 Bioanalyzer and using Qubit™ dsDNA HS Assay Kit (Thermo Fisher Scientific, NY, USA) on Qubit 2.0 Fluorometer (Thermo Fisher Scientific, NY, USA), respectively, according to manufacturer protocol. SMART-Seq® Stranded Kit is designed to analyze degraded, partially degraded or high-integrity RNA and deplete ribosomal cDNA. The constructed cDNA libraries were sequenced on an Illumina HiSeq 3000 sequencer as 50 base pair single-end reads at UCLA Technology Center for Genomics & Bioinformatics (TCGB). The Sus_scrofa and Homo_sapiens genome index were created using Sus_scrofa genome file (Sscrofa11.1.fa) and annotation file (Sus_scrofa.Sscrofa11.1.95.gtf), and Homo_-sapiens genome file (Homo_sapiens.GRCh38.dna. primary_assembly.fa) and annotation file (Homo_sapiens.GRCh38.97.gtf), respectively. These files were downloaded at https://uswest.ensembl.org /Sus_scrofa/Info/Index and https://uswest.ensembl.org/ Homo_sapiens/Info/Index. The fastq files from TCGB were then aligned against the genome assembly using STAR v2.7.1a[59], followed by assessment for the total number of aligned reads and total number of uniquely aligned reads to evaluate sequencing performance. The DEG lists were generated using edgeR in conjunction with Limma-Voom. Briefly, the gene counts were then imported into R package edgeR[60], a count-based statistical method, and trimmed mean of M-values (TMM) normalization size factors were calculated to adjust for differences in library size across samples. The TMM size factors and the matrix of counts were imported into the R package Limma, and a weighting approach with the voomWithQuality-Weights function and an additive generalized linear model in Limma were used to deal with variations in sample quality and to correct batch effects created by library preparation and sequencing of large numbers of samples over time, respectively[28]. An FDR–adjusted $p$ value threshold of $q < 0.05$ was used to subset meaningful genes, which were used for pathway enrichment analysis.

**scRNA-seq in colonic enteric ganglia of naive pigs**. The mucosa and submucosa were peeled off from each p-pC, p-tC and p-dC of 3 naive pigs (2 males and 1 female). The muscularis externa containing myenteric ganglia was processed for cell suspension preparation. Cell suspension preparation was performed according to the protocol presented by Smith et al.[61] with modifications. The original protocol aims to isolate a mixed population of neurons and glia from the myenteric plexus in mouse. The tissue samples were washed in ice-cold, carbogen (95% $O_2$ and 5% $CO_2$)-bubbled PBS. The muscularis externa containing myenteric ganglia were peeled off from the underlying tissue using forceps, followed by incubation in enteric neuron media, containing Neurobasal A media with B-27 (Thermo Fisher Scientific, NY, USA), 2 mM L-glutamine (Thermo Fisher Scientific, NY, USA), 1% fetal bovine serum (FBS) (Thermo Fisher Scientific, NY, USA), 10 ng/ml Glial Derived Neurotrophic Factor (Cedarlane Corporation, NC, USA) and 1× Anti-biotic/Antimycotic (Thermo Fisher Scientific, NY, USA), in the presence of 45 μM Actinomycin D (ActD)[59] (Thermo Fisher Scientific, NY, USA) for 15 min on ice. Application of ActD could minimize artificial transcriptional changes during tissue dissociation and enable unbiased characterization of cell types and their acute activation[62]. The samples were then cut into small pieces <1 mm and transferred to the enteric neuron media containing 1 mg/ml collagenase B (Sigma-Aldrich, MO, USA) and dispase II (Sigma-Aldrich, MO, USA) and 45 μM ActD for 1 hr at 37 °C. After the addition of 1 mg/ml deoxyribonuclease I (Sigma-Aldrich, MO, USA) and 10% FBS, the tissue pieces were dissociated using fire polished Pasteur pipettes. Following manual trituration, the cells were filtered through 70 μm Nitex mesh filter (Miltenyi Biotec Inc., CA, USA) and pelleted at 375 g for 10 min, and resuspended in the ice-cold, carbogen-bubbled rinse medium containing F12 media (Thermo Fisher Scientific, NY, USA) with 10% FBS and 1× antibiotic/antimycotic. Importantly, the solutions used in all steps were equilibrated in the carbogen gas. Estimation of viable cell number was done by Trypan Blue dye (Thermo Fisher Scientific, NY, USA) exclusion method. After cell staining with DAPI (Thermo Fisher Scientific, NY, USA), the viable cells were collected via fluorescence-activated cell sorting (FACS) (BD FACSAriaII, BD Biosciences, CA, USA) at UCLA Jonsson Comprehensive Cancer Center, whose viability was assessed using a Countess II FL Automated Cell Counter (Thermo Fisher Scientific, NY, USA) at UCLA TCGB. On average, the viable cells were collected via FACS accounted for around 80.4%. Before the cell suspensions were loaded on the 10× Genomics Chromium platform, Countess II FL Automated Cell Counter was used to select the cell suspensions with >70% viability. In order to determine the concentration of ActD, after cell suspension preparation from muscularis externa of porcine proximal colon, more than 1 million cells per sample were collected for bulk RNA-seq on an Illumina HiSeq 3000 sequencer and the same strategy (read alignment using

STAR and DEG list generation using edgeR) was applied to analyze the data. Immediate-early genes (IEGs) are rapidly and transiently induced by various cellular stimuli, which were chosen based on Wu et al.[62] The expression levels of total 127 IEGs were extracted (See Supplementary Fig 7). The heatmap was generated based on the average gene expression levels using R package 'heatmap.2'.

The scRNA-seq libraries were prepared from individual cells using the 10× Genomics platform. The Chromium Single Cell 3′ Library & Gel Bead Kit v2, Chromium Single Cell 3′ Chip kit v2 and Chromium i7 Multiplex Kit were used according to the manufacturer's instructions. On average, approximately 3200 cells from p-pC, 3800 cells from p-tC and 4100 cells from p-dC were loaded on the 10× Genomics Chromium platform and sequenced on an Illumina NextSeq 500 Sequencer (75 cycles). Sequencing was performed in paired-end mode. Cell Ranger 3.1.0 count function was used to align and quantify the reads against Sus_scrofa genome assembly, created using the Cell Ranger 3.1.0 mkref function with default parameters. This yielded an average of 1930, 2318 and 2559 mapped single cells (49884, 38516, and 43122 mean reads per cell) in p-pC, p-tC and p-dC, respectively.

**Identification of cell types from scRNA-seq data and analysis of ligand-receptor interaction in colonic enteric ganglia of naive pig.** The scRNA-seq data were processed for cell selection, filtration, normalization, scaling, cell cluster identification and annotation of neuronal and glial clusters using R toolkit Seurat. Briefly, after the selection and filtration of cells (genes expressed in at least 3 cells, cells with reads quantified for between 200 and 2500 genes and percentage of counts coming from mitochondrial genes less than or equal to 5%), data normalization and scaling were performed using default options. Data were batch-corrected using Seurat's integration method. The Seurat Integration first used canonical correlation analysis (CCA) to project the data into a subspace to identify correlations across datasets. The mutual nearest neighbors (MNNs) were then computed in the CCA subspace to correct the data. A K-nearest neighbor approach was employed to identify clusters using the top 10 principal components of the processed expression data with resolution set at 0.5. The UMAP algorithm was used for dimensionality reduction. All the dimensions of the low dimension representation were used as input to generate the UMAP plots with default parameters. Marker genes were identified by determining the average log-fold change of expression of each cluster compared to the rest of the cells using Seurat's FindAllMarkers function using the default settings for the Wilcoxon rank sum test. We identified marker genes as those with an average log-fold change above 0.25. Clusters were labeled using cell types associated with the identified marker genes. The neuronal and glial populations acquired from 9 scRNA-seq datasets were used to further gain the subpopulations of neurons and glia using Seurat. On average, 307, 172, and 266 neurons and 190, 191, and 233 glia were isolated from each p-pC, p-tC, and p-dC respectively, from which 3 neuronal (cholinergic, nitrergic and glutamatergic) and 2 glial subpopulations (marked by *SLC41A1* and *SPC24*, respectively) were identified. Cell-cell interactions were inferred from the single-cell transcriptomic data based on the ligand-receptor (L-R)-pair list in porcine generated using R package 'LRBase.Ssc.eg.db'. Firstly, scRNA-seq data were read using R package 'DropletUtils' and then the target cells were selected using unique molecular identifiers of neurons and glia. After data normalization, the R package 'scTGIF' was used to convert Ensembl ID to NCBI gene ID and a SingleCellExperiment object was created using R package 'SingleCellExperiment'. At last, the R package 'scTensor' was used to detect and visualize cell-cell interactions.

**Single-molecule fluorescence in situ hybridization (smFISH).** To validate the key discriminatory markers for the subsets of neuronal and glial cells clustered from scRNA-seq data, we performed RNAscope, a smFISH technique with fresh-frozen sections (12 μm) crossing the myenteric ganglia in p-pC. RNAScope Multiplex Fluorescent Kit v2 (Advanced Cell Diagnostics) was used per manufacturer's recommendations. The hybridization detection was carried out using Opal Fluorophore Reagent Packs Opal-520 (channel (C) 1)/570 (C2)/690 (C3) (1:1500) (Akoya Biosciences). Probes (Advanced Cell Diagnostics) used for smFISH include *GAP43* (1076491-C1) for annotating enteric neurons validated with *ELAVL4* (1076561-C2) encoding Hu C/D, a well-known pan neuronal marker; *CLDN11* (1076511-C1) for annotating enteric glial cells, *SLC41A1* (1076531-C1) and *SPC24* (1076541-C2) for annotating two subtypes of enteric glial cells, validated with *GFAP* (1039661-C3) encoding glial fibrillary acidic protein (GFAP), a well-known marker for pan enteric glial cells; *ACLY* (1076551-C2) for annotating cholinergic neurons validated with *ChAT* (849981-C1) encoding choline acetyltransferase (ChAT), a well-known marker for cholinergic neurons; *GLS* (1076571-C1) for annotating glutamatergic neurons validated with *SLC17A6* (1076581-C2) encoding vesicular glutamate transporter 2 (VGLU2), a well-known marker for glutamatergic neurons; *ARHGAP18* (107659-C1) for annotating nitrergic neurons validated with *NOS1* (1076601-C2) encoding neuronal nitric oxide synthase (nNOS or NOS1), a well-known marker for nitrergic neurons. A 3-plex RNAscope® positive control including 3 probes for pig *ACTB* (1076681-C1), *PPIB* (428591-C2) and *GAPDH* (1076691-C3), three housekeeping genes encoding β-actin (ACTB), peptidylprolyl isomerase B (PPIB) and glyceraldehyde 3-phosphate dehydrogenase (GAPDH), and a 3-plex negative control containing 3 probes with C1, C2, and C3 (Advanced Cell Diagnostics) were served to test the specificity of RNAscope assay and the qualities of tissue samples. Images were taken using a Zeiss LSM 710 confocal

microscope (Carl Zeiss Microscopy, LLC, White Plains, NY) with a laser set of 488 (C1)/561 (C2)/647 (C3). Five to ten optical sections (Z-stack) crossing the myenteric ganglia with frame 436 × 436 μm and 1 μm apart (20× objective) were acquired and overlaid using Imaris 9.7 for Neuroscientists (Bitplane Inc., Concord, MA).

**Deconvolution of bulk RNA-seq data using scRNA-seq datasets from colonic enteric ganglia of naive pig.** The R package 'SCDC' was used to deconvolve the bulk RNA-seq data from the myenteric ganglia of naive porcine colon with an ENSEMBLE method using default settings[24]. The 9 scRNA-seq datasets generated in this study were served as references. With marker genes specific for the specified cluster, five subpopulations of cells including cholinergic, nitrergic, glutamatergic neurons and two types of glia were identified in scRNA-seq datasets and selected to construct basis matrix in p-pC, p-tC and p-dC. For each scRNA-seq dataset with raw counts, a quality control procedure was carried out to filter the cells with the threshold of keeping a single cell from an assigned cluster of 0.7. The grid search method was used to derive the ENSEMBLE weights with the search step size of 0.01. SpearmanY from the grid search result represented the maximum of the Spearman correlation between observed cell-type gene expression in scRNA-seq datasets and predicted cell-type gene expression (probabilities) in bulk RNA-seq data. The cell-type gene lists were then matched to the DEG lists ($p < 0.05$) that were obtained from bulk RNA-seq data analysis. In this way, the cell-type DEG lists were generated and used for pathway enrichment analysis.

**Selection of orthologous genes.** A gene ortholog table was first established using human genome as the reference gene list to compare transcription between human and pig. Gene homology search was performed using ensemble multiple species comparison tool (http://www.ensembl.org/biomart/) to generate annotations for human-porcine orthologs. We first downloaded a list of orthologous genes from the ENSEMBL BioMart (release 104). We removed any duplicate annotations as well as any orthologs that were not strictly 1:1. A high-quality orthologous gene list was extracted with both gene order conservation score and whole genome alignment score above 75, resulting in 12291 genes. Details for calculation of whole genome alignment score can be found at https://uswest.ensembl.org/info/genome/compara/Ortholog_qc_manual.html#wga.

**Expression levels and cross-species normalization.** Based on the reads of the high-quality ortholog genes, the Python/bioinfokit (v0.9.1) package was used to calculate standard RPKM (reads per kilobase of exon model per million mapped reads), expression values for the orthologous gene set, followed by log₂ transformation:

$$\frac{\text{Number of reads mapped to gene} * 10^3 * 10^6}{\text{Total number of mapped reads} * \text{gene length in bp}} \quad (1)$$

The cross-species normalization was completed according to the scaling procedure reported by Brawand et al.[63]. Briefly, among the genes with expression values in the inner quartile range, the 8377 genes with the most conserved ranks among samples were identified and their median expression levels in each sample were assessed in R or Python without using any libraries. The scaling factors were derived by adjusting these medians to a common value and used to scale expression values of all genes in the samples.

The normalization was summarized as:

$$Normalized(e_i) = \frac{e_i \text{ in Sample } i}{\text{Median }(e_i) \text{ in Sample } i} * Average \ of \ Median(e_i) from \ all \ the \ samples \quad (2)$$

(Where $e_i$ is gene expression values in Sample $i$, $i$ = sample number).

After normalization, empirical cumulative density function (ECDF) (R/ecdf package) was used to estimate their probability distributions, and both Kolmogorov–Smirnov test (Python/scipy.stats.ks_2samp) and Pearson's Chi-squared test (Python/bioinfokit v0.9.5) were used to evaluate their goodness of fit. R/cor package was applied to compute the cross-species Spearman's Correlation, which was visualized using R/corrplot package.

**Analysis of pathway enrichment using bulk and single-cell RNA sequencing data.** The DEG lists were generated based on the transcriptomic comparisons in both human and naive pig samples with datasets pooled from males and females. Those from the male pigs were only used to assess the effects of VNS on the transcriptomic profiling in porcine colon. In addition to BPs used for the RNA-seq data interpretation, WikiPathways were chosen to provide intuitive views of the multiple interactions underlying BPs. We searched WikiPathways of interest from human database (https://www.wikipathways.org/) and downloaded all genes involved in the pathways to interpret the data from the porcine colon. The genes were matched to the porcine DEG lists that were already matched to the generated high-quality ortholog gene list. The resulting WikiPathways-matching DEG lists were used to perform enrichment analysis. WikiPathways of interest were classified into 9 categories (see details in Supplementary Table 3). Some WikiPathways such as smooth muscle contraction and activation of angiotensin pathway were considered to evaluate the influences of MG and ISG on the

physiological status of smooth muscle and mucosa. The WikiPathways such as gap junctions associated with electrical stimulation were also included. In addition, pro-/anti-inflammatory signaling (see details in Supplementary Table 4) was used to assess the effects of the vagal electrical stimulation on the inflammatory status. We used two methods (g:Profiler and ClueGO) to perform DEG-mediated pathway enrichment analyses, which led to the similar results. Firstly, using the genes that were detected in the bulk RNA-seq as the customized background, the DEG lists were imported into g:Profiler (https://biit.cs.ut.ee/gprofiler/) or ClueGO v.2.5.6, according to the protocol presented by Reimand et al.[29] and Bindea et al.[64]. ClueGO provides predefined selection criteria of representative pathways. We defined a term as specific (GO levels: 7–15) for one of the clusters if minimum genes/term were 1 and the mapped genes in the specific pathways represented more than 50% of the associated genes. For each DEG list, we selected the minimum GO level, where there were at least 3 BPs at an FDR $p$-value cut-off of 0.05. g:Profiler found the genes that were significantly enriched in BPs using a Fisher's exact test and multiple-test correction (Benjamini-Hochberg). The results from g:Profiler or ClueGO v.2.5.6 were inputted into EnrichmentMap v.3.2.1 or ClueGO v.2.5.6 in Cytoscape v.3.8.2[65] to visualize the functionally grouped networks of BPs with an FDR $p$-value cut-off of 0.05. We removed some annotations of BPs that were not correlative with ENS functions such as terms related to heart function. Edge width represented the correlation coefficient between nodes and the similarity cutoff overlap coefficient was set to 0.25. Each DEG from a WikiPathway-matching DEG list was searched in the grouped networks of BPs and all terms (BPs) containing the DEG were highlighted. After we obtained the significance (corrected $p$-value by Benjamini-Hochberg) of the highlighted terms related to each DEG, by treating the DEGs in the DEG list as dependent events, we combined the significance derived from each DEG by model averaging to compute the FDR $p$-value of the specified WikiPathway using R/harmonicmeanp package. The bubble plots were generated using R/ggplot2 package. Similarity matrix was generated in R/rrvgo package at default setting using DEG lists from comparison of p-pC-MG and p-pC-ISG in pigs with VNS. To further confirm whether the findings in pig have relevance to translational research, the porcine orthologous DEG list from human h-aC-MG vs h-tC-MG, h-aC-MG vs h-dC-MG or h-tC-MG vs h-dC-MG was imported into the EnrichmentMap that was established based on the regionally corresponding DEG list from porcine p-pC-MG vs p-tC-MG, p-pC-MG vs p-dC-MG or p-tC-MG vs p-dC-MG to assess the extent to which each two gene networks are overlapping. In the same way, gene network overlap was assessed based on the human orthologous DEG list from the regional comparisons in pig and the regionally corresponding DEG list from human or the regionally corresponding DEG lists from the pigs with and without VNS. R scripts used for data analyses in our study can be found in the public GitHub repositories (Supplementary Table 5).

**Defining the ortholog of human disease risk genes in the porcine colonic ENS.** The human disease risk genes relevant to Hirschsprung's disease, inflammatory bowel disease, autism spectrum disorders, and Parkinson's disease were selected based on the literatures[66–69]. Their orthologous genes were identified in the porcine colonic ENS and applied for genetic risk prediction of human diseases. Enrichment analysis of WikiPathways that regulate the expression of the disease risk genes was performed based on the above-mentioned procedure.

**Statistics and reproducibility.** Bulk RNA-seq data were collected from ISG and MG respectively of p-pC, p-tC and p-dC (6 without VNS and 3 with VNS) and from MG of h-aC, h-tC and h-dC (4 of each). Bulk RNA-seq data from 3 out of 6 animals without VNS were extracted to reflect the influences of VNS. Single-cell RNA-seq data were collected from the muscularis externa containing myenteric ganglia from each p-pC, p-tC and p-dC of 3 naive pigs. Therefore, at least three biological replicates for each experiment. We required replication at $p < 0.05$ using R packages. R scripts used for data analyses in our study can be found in the public GitHub repositories (Supplementary Table 5).

**Reporting summary.** Further information on research design is available in the Nature Portfolio Reporting Summary linked to this article.

## Data availability
RNA sequencing data have been deposited into the National Center for Biotechnology Information Gene Expression Omnibus under accession number: GSE197106. The data used to generate Fig. 2a were provided as Supplmenetary Data 1. The data used to generate Fig. 7 were presented as Supplementary Data 2. The data used to generated Fig. 8 were presented as Supplementary Data 3. The data used to generate the plots for Fig. 4c, d were provided as Supplementary Data 4–5, respectively.

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

## Acknowledgements

The present work was supported by the NIH SPARC OT2 grant OD024899 (P.Q.Y., Y.T., and M.M.), UCLA/Digestive Diseases Research Center Core P30 DK41301 and Animal Model Core (Y.T. and M.M.). We would like to thank Dr. Weizhe Hong, UCLA Department of Biological Chemistry and Department of Neurobiology for sharing Act-seq protocol and helpful comments. We thank UCLA Technology Center for Genomics & Bioinformatics (TCGB) for RNA sequencing service. We also like to acknowledge Dr. Bob Goldberg, UCLA Molecular, Cell, & Developmental Biology for providing LMD6000 facility.

## Author contributions

P.Q.Y. conceived and designed the project and performed experiments; T.L. designed, conducted major experiments and analyzed the data; M.M. (Mulugete Million) and M.L. supported the tissue collection and vagal nerve stimulation; M.M. (Marco Morselli) assisted in cDNA library construction; S.T. performed alignment for the reads from parts of bulk RNA-seq raw data; T.L., P.Q.Y. wrote the manuscript; Y.T., M.P., M.M. (Million Mulugeta), M.M. and S.T. gave input on the written manuscript; Y.T. supervised the project. All authors listed provide approval for publication.

## Competing interests

The authors declare no competing interests.
