## [Peer Review File · Communications Biology]

Reviewers' comments:

Reviewer #1 (Remarks to the Author):

The manuscript describes a number of comparisons of the transcriptomes of dissected components of the porcine and human enteric nervous system, as well as single cell RNA seq analysis using 10X chromium of porcine dissected tissues, to learn the pathways involved and different between anatomical locations and to compare human and pig. Further, some tissues were collected from animals with and without prior in vivo stimulation of the vagus nerve.

While many parts of this manuscript is expected to provide useful information for scientists interested in understanding gene expression in these anatomical regions, the experiments and analytical tools were not described sufficiently clearly so that an understanding of the analysis and the results could be performed. It is recommended that the section on DEG analysis and GO pathway analysis be rewritten (or re-analyzed depending on the response to the concerns below).

For example, on page five, it was unclear what authors meant by extraction of "functional linkages". There was no clear description of what such extraction means in the methods section. They apparently performed several ad hoc and poorly defined adjustments of both the differentially expressed gene list to remove potential contamination of other tissues and then apparently use this list in ClueGO. This is problematic as only the genes for which such an annotation is available will be removed, thus the remaining gene set is likely a mixture of genes involved in other pathways and some genes that are involved in pathways but are not completely/correctly annotated. It is recommended to leave all genes in the analysis and the GO annotation will provide an appropriate result. As well, in a second method: they also used only some differentially expressed genes "involved in the WikiPathways of interest" to "perform enrichment analysis". To identify enrichment of function within DEG, normally all DEG are used and the frequency of a GO term is tested against an appropriate background (one suggestion would be to use as background all genes detected in the samples under study, as this list comprises those genes that could have been predicted to be DE). The authors did not justify selecting genes within a pathway first, and thus it is hard to see how enrichment is appropriately calculated.

In figure 1D, what the nodes in the diagram are representing is not clear- are these distinct WikiPathways or are they substantially the same pathways with slightly different annotation? In the upper right, it appears there are 10 nodes which are represented by three porcine orthologs: how is it possible to represent 10 pathways with just three genes? Even if this can be clarified, the diagrams in 1D appear to show almost no overlap in differentially expressed genes between porcine and human DEG lists. As one of the conclusions of the manuscript are that porcine and human pathways are similar enough that the pig can be a good model for human anatomical differences, this needs to be carefully described. If the figure is correct, only three to 4% of the genes in human overlap with pig, as seen in figure 1D, top left, and only one to 2% overlap in 1D, top right.

At another level, it is really unclear that the overlaps are limited to these "porcine orthologues" since a) these are also part of the human gene list as well, (correct?), and b) are they saying there are no other poor sign orthologues in the human list?? It is hard to believe this since many thousands of genes are orthologous between human and pig (as the authors indicate as well, page 22).

These concerns are important since the number of differentially expressed genes that predict an enriched GO term is crucial for interpretation. It appears they used as few as three (or even one!) genes to declare a biological process pathway is significantly different between samples (Suppl. Note 1). This is weak evidence for such a declaration, especially since they use such pathways to interpret results for much of the rest of the paper. A table showing the number of genes predicting such enrichment at the GO term level would be useful for all comparisons.

This clarification is most important because it is critical to the main conclusion of the first section as well as much of the rest of the paper. As the authors declare these very few orthologs have functional linkages that represent all the characteristics of the porcine posterior/distal colon for the mesenteric ganglion (bottom page 5). Nothing was provided to really validate this statement.

Further, since figures two and three are illustrations of the results from these methods, it was not possible to really evaluate these figures and the relevant supplemental figures. A similar comment can be made for several other figures, including 6,7,8 and 9.

The single cell RNA seq analysis appears to have been performed appropriately with correct quality control and comparisons using Seurat tools. However, in Figure 4A and B, the middle panel shows apparently expression of 11 genes, yet only two are labeled; this omission is for no apparent reason and the missing genes are not described anywhere. If trying to reduce clutter in this figure, you could list the genes in order in the legend, although that would not be recommended as clearly these genes are very useful for annotating the other clusters. In Figure 5, there is good corroboration of many of these markers. Although there was some labeling errors (in 5B Hu is label (?)) but no such gene is described). While some patterns of pairs of genes tested do look coincident, for example the A-B pair the J-K pair and the D-E pair, some did not. For example, the G-H pair and the M-N pair. It is unclear why discussion of Figure 6 follows the single cell RNA seq work; there was no explicit mention of the single cell RNAseq data in this paragraph.

Reviewer #2 (Remarks to the Author):

In this work, Li et al., have profiled the transcriptomes of myenteric and inner submucosal ganglia from porcine proximal, transverse and distal colon (p-pC, p-tC, p-dC), and MG and compared it with human analogous regions using several transcriptomics techniques and validated some of their results using additional methods, such as sm-FISH.

The data presented seems as an important resource to compare porcine with human colon and the analyses made are comprehensive. However, I find much of the text unclear and many details that are important for understanding the analysis and for reproducing the results are missing.

My general comments are:

1) While the introduction is written in a very clear manner, the Results are not clear and start immediately with a very specific and unclear paragraph. The authors should start by introducing the system and the analytical procedures. Furthermore, many sentences in the results should be simplified and broken into several shorter sentences. For example, the Results starts with the following: " The obtained empirical cumulative density function (ECDF) profiles and both Kolmogorov-Smirnov and Pearson's Chi-squared test ($p > 0.05$) based on the averaging normalized transcription levels of each orthologous gene demonstrated that gene expression profiling in MG of human colonic segments (h-aC, htC, h-dC) could be predicted according to that of corresponding porcine colonic segments (p-pC, p-tC, pdC), following the probability distribution shown in ECDF profiles (Fig. 1a, c)."

2) In addition, a figure that describes the study in terms of overall design, including main features (regions, species, condition, number of biological replicates, number of cells, methods) is missing and should appear as Fig 1A.

3) The computational analysis will be more replicable by others, increasing the impact of this work, if the authors provide an online notebook / important scripts in a depository such as GitHub.

Specific comments:

1) Continuing point #3 above, the entire procedure of the ortholog expression comparison should be accompanied with a detailed script as it is the basis of many analyses and is not clear from the short text provided by the authors:

"Based on the reads of the high-quality ortholog genes, the Python/bioinfokit (v0.9.1) package was used to calculate standard RPKM (reads per kilobase of exon model per million mapped reads) expression values for the orthologous gene set, followed by log₂ transformation. The cross-species normalization was completed according to the scaling procedure reported by Brawand et al.⁵⁷."

2) Fig 1b – the scale (-1 to +1) used is too large – if all correlations are positive (And it is difficult

to say, but it looks above 0.4?) – you should use a much higher value as the min.

Also, all P-values are given as three stars, and with the current colors, it is difficult to assess whether the similarity between analogous regions between the species is higher – for example is p-dC-MG more highly correlated with h-dC-MG than h-tC-MG or h-aC-MG?

3) Fig 1d – I could not follow the authors in the text related to this figure panel, and the figure itself is not simple to understand:

The paragraph that starts with: “The differentially expressed genes (DEG) enriched significantly ($p < 0.05$) by Gene Ontology terms were selected for comparisons among colonic segments, between MG and ISG, and the porcine and human.” Should be rephrased and the message the authors are trying to convey should be written in a simpler and clearer way: The sentences are often too long, and some of them (e.g. “Twelve porcine orthologous DEG (9 upregulated and 3 downregulated) in comparison of h-aCMG and h-dC-MG were involved in the regulatory networks, mediated by 483 DEG (271 upregulated and 212 downregulated), when comparing p-pC-MG and p-dC-MG (Fig. 1d).”) do not make much sense.

4) In the statement “We did not find any porcine orthologous DEG when comparing haC-MG or h-dC-MG with h-tC-MG.” - what do you mean by not finding any ortholog? Is there no 1-to-1 ortholog? Do you not have any gene below a certain cutoff of the DE analysis?

5) In Methods, in Page 20:

All of the reads were then aligned to the *sus*scrofa or human genome using STAR and the DEG lists were generated using edgeR (Supplementary method 3) and used for pathway enrichment analysis.

Please add details with respect to which genome assemblies were used and from what source they were taken from. This should appear in Main, not in Supp.

Notice that in Supp you write: “Susscrofa genome files (.fa) and annotation file (.gtf) were downloaded at <https://uswest.ensembl.org/Susscrofa/Info/Index>, which were used to create the susscrofa genome index...” – this has no mention of the pig’s genome version.

6) Also, add details of the edgeR analysis to Main text – which genes were taken into account in the DE analysis, which test you used, etc.

7) In “Selection of orthologous genes” – it is not entirely clear how do you select your orthologs: do you use ENSEMBL orthology annotations between human and pig? If so, do you only use one-to-one orthologs? If you use one-to-many / many-to-many orthologs as well, how do you treat the “many”? e.g. -do you sum their expression together?

Which ENSEMBL biomart version did you use?

What do you mean by “whole genome alignment score above 75”? the reference given (#56) is probably the wrong reference as it seems unrelated. By “whole genome alignment” do you mean the entire gene body or just the CDS? Or the transcripts?

Please add all these details to Methods.

8) In Supplementary Method 4 – cell suspension preparation – have the authors tried or considered different dissociation protocols? Why was this protocol used? Was there an attempt to enrich for a certain population of cells? What may be the biases and caveats in the protocol used? How many viable cells did you usually get following dissociation (based on either Trypan blue & microscopy and FACS)?

9) In Supplementary Method 5 – you should mention the specific batch correction method used (Seurat has several such methods). Similarly, in :The UMAP algorithm was used for dimensionality reduction.” Should include more details.

10) In Fig 4 – how many cells and from how many individuals appear in panels A and B? (please add number of cells and individuals to Fig Legend)

In methods you write that for single-cell, you used 4 individuals and loaded ~10,000 single cells per cell suspension. Can the same figures, colored by individual, be shown in Supp as well?

Minor comments:

1) In Page 6, In:

"The interactions were defined when the networks mediated by DEG involved in WikiPathways of interest and Biological Processes (BPs) shared BPs."

I guess that by "interactions" you refer to "cross-species similarities"? please rephrase.

And, later in text – what do you mean by "Innate similarities"?

2) In page 24, In:

"Their orthologous genes were defined in the porcine colonic ENS and applied for genetic risk prediction of human diseases"

You probably mean "identified"

3) In Discussion, In:

"The networks mediated by the human orthologous genes from comparison of p-pC-MG and p-dC-MG did not realize 100% coverage of the linkages derived from the DEG lists in comparison of h-aC-MG and h-dC-MG"

What do you mean by "did not realize"?

We thank the reviewers for the careful and insightful review of our manuscript and thorough and constructive comments towards improving our manuscript. All comments were taken into consideration and addressed accordingly in the revised version. A point-by-point reply to reviewers' comments is as follows.

Reviewer # 1

Comment 1. *It is recommended that the section on DEG analysis and GO pathway analysis be rewritten (or re-analyzed depending on the response to the concerns below).*

Reply: As recommended the DEG analysis and GO pathway analysis have been rewritten according to the following comments (see Page 5, 6, 7, 9, 21, 23, 24, 25 and 26 and see Supplementary Method 5 and Supplementary Note 1).

Comment 2 *Page five, it was unclear what authors meant by extraction of “functional linkages”. There was no clear description of what such extraction means in the methods section. They apparently performed several ad hoc and poorly defined adjustments of both the differentially expressed gene list to remove potential contamination of other tissues and then apparently use this list in ClueGO. This is problematic as only the genes for which such an annotation is available will be removed, thus the remaining gene set is likely a mixture of genes involved in other pathways and some genes that are involved in pathways but are not completely/correctly annotated. It is recommended to leave all genes in the analysis and the GO annotation will provide an appropriate result.*

Reply:

- Extraction of functional linkages is now described in the result section (Line 125-128 in Page 6) as follows “The orthologous DEG were searched in the grouped networks of GOBP terms and all terms containing the DEG was highlighted. Consequently, the information for the DEG-enriched terms and the number of genes shared by the connected terms (functional linkages) was used for comparisons of functional differences.”
- We thank the reviewer for pointing to the potential biases of using our method to remove potential contamination of other tissues. We now follow the recommendation to leave all genes in the analysis. Actually, in our original analyses, we did not find any unique smooth muscle marker genes derived from scRNA-seq in each DEG list based on bulk RNA-seq data. The statement was added to prove that we have considered the scenarios where there was the potential contamination from intestinal muscle during LCM and have applied our computational methods to eliminate the potential contamination. In the method section, we deleted the statement “In order to remove the potential contamination from intestinal muscle during LCM, unique smooth muscle marker genes derived from scRNA-seq were deleted from each DEG list based on bulk RNA-seq data”. (Page 25)

Comment 3. *In a second method, they also used only some differentially expressed genes “involved in the WikiPathways of interest” to “perform enrichment analysis”. To identify enrichment of function within DEG, normally all DEG are used and the frequency of a GO term is tested against an appropriate background (one suggestion would be to use as background all genes detected in the samples under study, as this list comprises those genes that could have been predicted to be DE). The authors did not justify selecting genes within a pathway first, and thus it is hard to see how enrichment is appropriately calculated.*

Reply: This is an important consideration related to our strategy of performing enrichment analysis. We think that a lack of detailed description in the method section may lead to the

misunderstanding. We provided more details of pathway enrichment analysis in the method section: “We searched WikiPathways of interest from human database (<https://www.wikipathways.org/>) and downloaded all genes involved in the pathways to interpret the data from the porcine colon. The genes were matched to the porcine DEG lists that were already matched to the generated high-quality ortholog gene list. The resulting WikiPathways-matching DEG lists were used to perform enrichment analysis as described below.....

We used two methods (g:Profiler and ClueGO) to perform DEG analyses, which led to the similar results. Firstly, the DEG lists were imported into g:Profiler (<https://biit.cs.ut.ee/gprofiler/>) or ClueGO v.2.5.6, followed by creation and visualization of the functionally grouped networks of BPs in EnrichmentMap v.3.2.1 or ClueGO v.2.5.6 in Cytoscape v.3.8.2 with an FDR p-value cut-off of 0.05. The WikiPathways-matching DEG lists were searched in the grouped networks of BPs and all terms (BPs) containing the DEG were highlighted. The significance (corrected p-value by Benjamini-Hochberg) of the highlighted terms were exported to compute the FDR p-value of the specified WikiPathways using R/harmonicmeanp package.” (Line 601-605 and Line 611-618 in Page 25 and 26)

Comment 4

a) *In figure 1D, what the nodes in the diagram are representing is not clear- are these distinct WikiPathways or are they substantially the same pathways with slightly different annotation?*

Reply: The nodes in the diagram are clarified in the result section and the figure legend. For example, we added the statement “the orthologous DEG were searched in the grouped networks of GOBP terms and all terms containing the DEG was highlighted ...” and “... These 12 porcine orthologous DEG were then searched in the grouped networks of Biological Processes (BPs) (nodes shown in Fig. 2d)” to the result section (Line 125-127 and Line 132-133 in Page 6) and detailed the legend by addition of “Gene Ontology Biological Processes terms (pathways) are shown as circles (nodes) that are connected with lines (edges) if the terms share many genes. Nodes are colored in purple (without VNS) and in light green (with VNS), and edges are sized on the basis of the number of genes shared by the connected pathways” (Line 813-816 in Page 34).

b) *In the upper right, it appears there are 10 nodes which are represented by three porcine orthologs: how is it possible to represent 10 pathways with just three genes? Even if this can be clarified, the diagrams in 1D appear to show almost no overlap in differentially expressed genes between porcine and human DEG lists.*

Reply:

- In the reply to Comment 2, we added new descriptions “The orthologous DEG were searched in the grouped networks of GOBP terms and all terms containing the DEG was highlighted. Consequently, the information for the DEG-enriched terms and the number of genes shared by the connected terms (functional linkages) was used for comparisons of functional differences.” in the result section (Line 125-128 in Page 6). We updated figure 2D (original figure 1D) and the number of the nodes (Biological Processes) containing the porcine orthologs was specified. We do not have evidence to clarify whether the small number of the porcine orthologs can represent about 60 pathways. However, we addressed this question by focusing on the gene networks (functional linkages related to the porcine orthologs), which can represent the dynamic properties underlying gene interactions at a systems level and provide a large-scale view of the physiological state at the mRNA level. In both result and discussion section, we mentioned that functional linkages (genes shared by the connected pathways) related to the porcine orthologous DEG achieved 100% linkage coverage in the regulatory networks, mediated by the DEG when comparing p-pC-MG and p-dC-MG. From this perspective, we think (Line 308-309 in the discussion section, Page 13) the

porcine orthologous genes serve as driver genes, whose networks represent all the characteristics of p-pC-MG and p-dC-MG in porcine, in terms of 100% linkage coverage.

- We've noticed that there is almost no overlap in differentially expressed genes between porcine and human DEG lists, probably due to the lack of complete annotation and assembly of the porcine genome. After all of the reads were aligned to the Sus_Scrofa or human genome using STAR, we got 24358 and 60617 annotated genes in porcine and human datasets, respectively.

c) As one of the conclusions of the manuscript are that porcine and human pathways are similar enough that the pig can be a good model for human anatomical differences, this needs to be carefully described. If the figure is correct, only three to 4% of the genes in human overlap with pig, as seen in figure 1D, top left, and only one to 2% overlap in 1D, top right. .

Reply:

- In the body of the manuscript, we did not mention “the pig can be a good model for human anatomical differences because of the similar enough pathways in porcine and human”. We just demonstrated that “although there were few conserved genes between porcine and human colon, the **networks of interacting orthologous genes** orchestrated almost the same complex colonic functions across species, and p-pC and p-dC could serve as **transcriptional predictors**, with **important implications in understanding** of human colonic functions and neuromodulation.” (Line 314-317 in Page 13, the discussion section).
- Regarding Fig. 2D (original Fig. 1D), there are two possible reasons for less overlapped genes: 1) The porcine genome lacks complete annotation and assembly. After all of the reads were aligned to the Sus_Scrofa or human genome using STAR, we got 24358 and 60617 annotated genes in porcine and human datasets, respectively; 2) We cannot figure out whether the DEG in human and pig are involved in divergent or conserved pathways because we do not have evidence to clarify whether the small number of the porcine orthologs can represent about 60 pathways (updated figure 2D). Even if they are involved in some divergent pathways or a single pathway involved different human and porcine DEG, we can still understand the scenarios because there are species-specific transcriptional features. However, the porcine orthologous genes eventually orchestrate the same gene networks, implying almost the same complex colonic functions across species. Therefore, from a large-scale view, gene networks are conserved between human and porcine. Conservation is based on the gene regulatory networks and we stick to the principle in the following analyses to draw our conclusions.

Comment 5. *it is really unclear that the overlaps are limited to these “porcine orthologues” since a) these are also part of the human gene list as well, (correct?), and b) are they saying there are no other poor sign orthologues in the human list?? It is hard to believe this since many thousands of genes are orthologous between human and pig (as the authors indicate as well, page 22).*

Reply: The “porcine orthologues” are parts of the human gene list and there are poor sign orthologues in the human list. In the method section, we described “A high-quality orthologous gene list was extracted with both gene order conservation score and whole genome alignment score above 75, resulting in 12291 genes” (Line 576-578 in Page 24). The alignment score was selected based on the literature (Geirsdottir et al., Cell, 2019). The reason why we extracted the high-quality orthologous gene list is that the orthologous genes with high whole genome alignment score often have the same function in the different species. Then, we “matched the DEG lists to the high-quality ortholog gene list” (Line 603-604 in Page 25). The gene networks mediated by these “porcine orthologues” and all the human DEG in the same comparison are the same, suggesting that these

“porcine orthologues” could be the representatives to indicate overall similar functions across species. These are the reasons why we focus on the overlaps that are limited to these “porcine orthologues”. We added the statement “The functional linkages related to all DEG in comparison of h-aC-MG and h-dC-MG were the same as those related to the orthologous DEG (data not shown)” in the result section (Line 136-137 in Page 6).

Comment 6. *These concerns are important since the number of differentially expressed genes that predict an enriched GO term is crucial for interpretation. It appears they used as few as three (or even one!) genes to declare a biological process pathway is significantly different between samples (Suppl. Note 1). This is weak evidence for such a declaration, especially since they use such pathways to interpret results for much of the rest of the paper. A table showing the number of genes predicting such enrichment at the GO term level would be useful for all comparisons. This clarification is most important because it is critical to the main conclusion of the first section as well as much of the rest of the paper.*

Reply:

- As we mentioned previously, based on all DEG derived from each comparison, two methods were used (g:Profiler and ClueGO) to create and visualize the functionally grouped networks of the enriched GO terms (Biological Processes) (Line 611-614 in the method section, Page 25). This means there are no any biases to interpret the data.
- Next, the orthologous or WikiPathways-matching DEG were searched in the created GO term networks to find their directly involved GO terms and functional linkages (genes shared by the connected pathways), which were used for the downstream analyses (Line 125-128 and Line 132-134 in the result section, Page 6; Line 614-618 in the method section, Page 25 and 26).
- In the first section (cross-species comparisons), the orthologous DEG were selected and searched in the corresponding GO term networks. One hundred percent of linkage coverage indicates the similar gene networks and consequent similar functions across species (as mentioned in Comment 4). The conclusion was supported by the results from ECDF profiles, Kolmogorov-Smirnov and Pearson's Chi-squared test and Spearman's correlation (Line 106-120 in the result section, Page 5). Therefore, the porcine orthologous DEG could reflect functional features in the human colon. In addition, considering that the gene networks mediated by the porcine orthologous DEG conserved >90% of the transcriptomic responses to VNS, the orthologous DEG could be used for the prediction of performance in human neuromodulation (Related to Line 137-140 in the result section, Page 6; Line 292-296 in the discussion section, Page 12 and 13). In the following analyses, we mainly focused on the orthologous DEG.
- In the method section (Line 601-605 in Page 25), we mentioned that “We searched WikiPathways of interest from human database (<https://www.wikipathways.org/>) and downloaded all genes involved in the pathways to interpret the data from the porcine colon. The genes were matched to the porcine DEG lists that were already matched to the generated high-quality ortholog gene list. The resulting WikiPathways-matching DEG lists were used to perform enrichment analysis as described below.....”. Actually, in the result section (Line 148-149 in Page 7), we indicated that “The functional linkages related to each DEG list in our study and those related to the DEG list screened by the high-quality human orthologous genes achieved almost 100% linkage coverage”. We just compared the same WikiPathways for each comparison. This could avoid the bias of incomplete annotation in pathways. We used the same methods to search WikiPathways-matching DEG, extract the

directly involved GO terms and calculate the combined p -value. These could solidify our conclusions in this study.

- Based on all DEG derived from each comparison, we used two methods (g:Profiler and ClueGO) to create and visualize the functionally grouped networks of the enriched GO terms, which led to the similar results (Related to Line 611-612 in the method section, Page 25). We added more details for predefined selection criteria of representative pathways in Suppl. Note 1 (Line 94-97 in Suppl.). We defined a term as specific (GO levels: 7-15) for one of the clusters if minimum genes/term were 1 and the mapped genes in the specific pathways represented more than 50% of the associated genes. For each DEG list, we selected the minimum GO level, where there were at least 3 BPs at an FDR p -value cut-off of 0.05.

Comment 7. *As the authors declare these very few orthologs have functional linkages that represent all the characteristics of the porcine posterior/distal colon for the mesenteric ganglion (bottom page 5). Nothing was provided to really validate this statement. This clarification is most important because it is critical to the main conclusion of the first section as well as much of the rest of the paper.*

Reply: We defined the functional linkages as the number of genes shared by the connected pathways (Line 125-128 in the result section, Page 6). They represent gene networks. We mentioned that “The functional linkages related to the orthologous DEG achieved 100% linkage coverage” (Line 134-136 in the result section, Page 6). This means that they share the same gene networks. From this perspective, we demonstrated that “the 12 orthologous genes served as driver genes, whose networks represented all the characteristics of p-pC-MG and p-dC-MG in porcine, in terms of 100% linkage coverage” (Line 308-309 in the discussion section, Page 13).

Comment 8. *Further, since figures two and three are illustrations of the results from these methods, it was not possible to really evaluate these figures and the relevant supplemental figures. A similar comment can be made for several other figures, including 6,7,8 and 9.*

Reply: We added some details in the result and method section (e.g. Line 125-128 and Line 132-134 in Page 6 and Line 601-605 in Page 25). Please see the above-mentioned replies.

Comment 9.

a) *The single cell RNA seq analysis appears to have been performed appropriately with correct quality control and comparisons using Seurat tools. However, in Figure 4A and B, the middle panel shows apparently expression of 11 genes, yet only two are labeled; this omission is for no apparent reason and the missing genes are not described anywhere. If trying to reduce clutter in this figure, you could list the genes in order in the legend, although that would not be recommended as clearly these genes are very useful for annotating the other clusters.*

Reply: Thank you for the positive comment on the single cell RNA seq. We re-generated the Figure 4A and B (Line 830 in Page 37). In the Figures, we labeled all the cell populations based on all the listed markers. In the legend, we mentioned that “The clusters were labeled according to their markers and only neuronal and glial cell markers were verified by in situ hybridization” (Line 832-833 in Page 37).

b) *In Figure 5, there is good corroboration of many of these markers. Although there were some labeling errors (in 5B Hu is label (?) but no such gene is described). While some patterns of pairs of genes tested do look coincident, for example the A-B pair the J-K pair and the D-E pair, some did not. For example, the G-H pair and the M-N pair.*

Reply: We appreciated the careful review and apologized for the labeling errors. We replaced “Hu” with “ELAVL4”, “VGLU2” with “SLC17A6” and “SLC41A” with “SLC41A1” in the new figure (Line 848 in Page 38). In addition, we re-did RNAscope in situ hybridization for two pairs of marker genes ARHGAP18/NOS1 and CLDN11/GFAP. New images showed that the signals of ARHGAP18 (G) and NOS1 (H) or CLDN11 (M) and GFAP (N) were exactly overlapped in the merged images (I or O). The Figs G, H, I and M, N, O were updated with new data accordingly.

c) It is unclear why discussion of Figure 6 follows the single cell RNA seq work; there was no explicit mention of the single cell RNAseq data in this paragraph.

Reply: To address this comment, we added sentences “The resulting cell-type gene markers were then matched to the DEG lists from bulk RNA-seq data analysis. Consequently, the created cell-type DEG lists were used for pathway enrichment analysis” in the result section (Line 205-206 in Page 9). We also mentioned that the WikiPathways with top ranking enrichment scores in comparison of p-pC-MG and p-dC-MG were selected to evaluate contribution of each cell subset to pathway enrichment.” (Line 206-208 in the result section, Page 9).

Reviewer #2

General comments:

Comment 1. *the Results are not clear and start immediately with a very specific and unclear paragraph. The authors should start by introducing the system and the analytical procedures. Furthermore, many sentences in the results should be simplified and broken into several shorter sentences. For example, the Results starts with the following: “ The obtained empirical cumulative density function (ECDF) profiles and both Kolmogorov-Smirnov and Pearson's Chi-squared test ($p > 0.05$) based on the averaging normalized transcription levels of each orthologous gene demonstrated that gene expression profiling in MG of human colonic segments (h-aC, htC, h-dC) could be predicted according to that of corresponding porcine colonic segments (p-pC, p-tC, pdC), following the probability distribution shown in ECDF profiles (Fig. 1a, c).”*

Reply: Thanks reviewer's suggestions. At the end of the introduction section (Line 93-103 in Page 4 and 5), we had introduced our system and the analytical procedures and also added a new figure to describe the study in terms of overall design (Fig. 1). In the revised version, we added several sentences at the beginning of the results to explain why we used these methods: “We cross-compared the corresponding regional transcriptional profiling of MG between porcine and human to determine to what extent porcine model offers access to the colonic translational research based on the transcriptomic similarities. A possible explanation for those similarities was gleaned from comparisons of empirical cumulative density function (ECDF) profiles of the averaging normalized transcription levels of each high-quality orthologous gene. Both Kolmogorov-Smirnov and Pearson's Chi-squared test were applied to quantify a distance between ECDF profiles. Results demonstrated high similarities of probability distributions of human-porcine regional gene expression profiling in MG between corresponding colonic segments in the pig and human (p-pC vs h-aC, p-tC vs h-tC and p-dC vs h-dC). This was supported by each pair of ECDF plots being almost merged together and the lack of significant difference between the pair of ECDF profiles ($p > 0.05$), assessed by Kolmogorov-Smirnov and Pearson's Chi-squared test (Fig. 2a, c). Thus, the human colonic segments (h-aC, h-tC, h-dC) could be predicted according to that of corresponding porcine colonic segments.” (Line 106-117 in Page 5). In addition, we have broken the sentences into several shorter sentences.

Comment 2. *A figure that describes the study in terms of overall design, including main features*

(regions, species, condition, number of biological replicates, number of cells, methods) is missing and should appear as Fig 1A.

Reply: We added a new figure (Fig. 1) that describes the study in terms of overall design, including main features (regions, species, condition, number of biological replicates, methods) (Line 796-797 in Page 32). The number of cells that used for cluster identification was added in the method section (Line 520 in Page 22). A sentence “On average, the final number of the cells used for cluster identification in each sample was ~2500.” was added.

Comment 3. *The computational analysis will be more replicable by others, increasing the impact of this work, if the authors provide an online notebook / important scripts in a depository such as GitHub.*

Reply: Thanks reviewer’s comments. The documentation regarding the packages (R or Python) that we used can be found online, where gives much more details. In the method sections, we detailed the analysis procedures and the parameters that we changed (e.g. Line 559-568 in Page 23 and 24). These should help authors to easily follow our computational analysis.

Specific comments:

1. *Continuing point #3 above, the entire procedure of the ortholog expression comparison should be accompanied with a detailed script as it is the basis of many analyses and is not clear from the short text provided by the authors: “Based on the reads of the high-quality ortholog genes, the Python/bioinfokit (v0.9.1) package was used to calculate standard RPKM (reads per kilobase of exon model per million mapped reads) expression values for the orthologous gene set, followed by log2 transformation. The cross-species normalization was completed according to the scaling procedure reported by Brawand et al.57.”*

Reply: Thank you for reviewer’s comments. The R or Python packages used here are very simple without any complex parameters that need to be adjusted. In addition, the users can find very detailed documentation for the R or Python packages. In order to make the entire procedure clearer, we added two formula for: 1. RPKM calculation.
$$\frac{\text{Number of reads mapped to gene} \times 10^3 \times 10^6}{\text{Total number of mapped reads} \times \text{gene length in bp}}$$
 (see Line 582-584 in Page 24); 2. normalization procedure. The normalization was summarized as
$$\text{Normalized } (e_i) = \frac{e_i \text{ in Sample } i}{\text{Median } (e_i) \text{ in Sample } i} * \text{Average of Median } (e_i) \text{ from all the samples}$$
 (Where e_i is gene expression values in Sample i , i = sample number). (see Line 589-591 in Page 24).

2. *Fig 1b – the scale (-1 to +1) used is too large – if all correlations are positive (And it is difficult to say, but it looks above 0.4?) – you should use a much higher value as the min. Also, all P-values are given as three stars, and with the current colors, it is difficult to assess whether the similarity between analogous regions between the species is higher – for example is p-dC-MG more highly correlated with h-dC-MG than h-tC-MG or h-aC-MG?*

Reply: We updated the Fig 2b (original Fig 1b, Line 807 in Page 33) and made following changes: 1. The range of the correlation coefficients (r) was scaled from 0.7 to 1; 2. Each individual r value was added in the graph to show the correlation strength in corresponding regions between two species. All p-values are equal or less than 0.001 and marked as three asterisks, indicating highly significant correlations; 3. New sentences were added in the text as “pair-wise comparison showed a significant Spearman’s correlation in the corresponding regions between two species ($p < 0.001$) (Fig. 2b), suggesting the transcriptomic similarities of analogous MG in the corresponding colonic segments between pig and human.” (Line 117-120 in Page 5).

3. Fig 1d – I could not follow the authors in the text related to this figure panel, and the figure itself is not simple to understand: The paragraph that starts with: “The differentially expressed genes (DEG) enriched significantly ($p < 0.05$) by Gene Ontology terms were selected for comparisons among colonic segments, between MG and ISG, and the porcine and human.” Should be rephrased and the message the authors are trying to convey should be written in a simpler and clearer way: The sentences are often too long, and some of them (e.g. “Twelve porcine orthologous DEG (9 upregulated and 3 downregulated) in comparison of h-aCMG and h-dC-MG were involved in the regulatory networks, mediated by 483 DEG (271 upregulated and 212 downregulated), when comparing p-pC-MG and p-d. 1d).”) do not make much sense.

Reply: We updated Fig 2d (original Fig 1d, Line 808 in Page 34) and detailed the legend by addition of “Gene Ontology Biological Processes terms (pathways) are shown as circles (nodes) that are connected with lines (edges) if the terms share many genes. Nodes are colored in purple (without VNS) and in light green (with VNS), and edges are sized on the basis of the number of genes shared by the connected pathways” (Line 813-816 in Page 34). We rewrote this part and added more details to help others to understand our analysis procedures. Please see the revision in Line 121-145 in Page 6.

4. In the statement “We did not find any porcine orthologous DEG when comparing haC-MG or h-dC-MG with h-tC-MG.” - what do you mean by not finding any ortholog? Is there no 1-to-1 ortholog? Do you not have any gene below a certain cutoff of the DE analysis?

Reply: We did find the genes below a certain cutoff of the DE analysis. However, when we matched the DEG to the high-quality orthologous gene list, we did not find any porcine orthologous DEG (1-to-1 ortholog) when comparing h-aC-MG or h-dC-MG with h-tC-MG. We changed the statement to “The DEG were matched to the high-quality orthologous gene list, but we did not find any porcine orthologous DEG when comparing h-aC-MG or h-dC-MG with h-tC-MG.” (Line 142-144 in the result section, Page 6)

5. In Methods, in Page 20: All of the reads were then aligned to the suscrofa or human genome using STAR and the DEG lists were generated using edgeR (Supplementary method 3) and used for pathway enrichment analysis. Please add details with respect to which genome assemblies were used and from what source they were taken from. This should appear in Main, not in Supp. Notice that in Supp you write: “Suscrofa genome files (.fa) and annotation file (.gtf) were downloaded at <https://uswest.ensembl.org/Suscrofa/Info/Index>, which were used to create the suscrofa genome index...” – this has no mention of the pig’s genome version.

Reply: Thanks for reviewer’s suggestion. We added the details in the main method section. We mentioned “The Sus_scrofa and Homo_sapiens genome index were created using Sus_scrofa genome file (Sscrofa11.1.fa) and annotation file (Sus_scrofa.Sscrofa11.1.95.gtf), and Homo_sapiens genome file (Homo_sapiens.GRCh38.dna.primary_assembly.fa) and annotation file (Homo_sapiens.GRCh38.97.gtf), respectively. These files were downloaded at https://uswest.ensembl.org/Sus_scrofa/Info/Index and https://uswest.ensembl.org/Homo_sapiens/Info/Index.” (Line 501-505 in Page 21)

6. Also, add details of the edgeR analysis to Main text – which genes were taken into account in the DE analysis, which test you used, etc.

Reply: We thank reviewer’s suggestion. The details of the edgeR analysis were added in the main method section as “The DEG lists were generated using edgeR in conjunction with Limma-Voom. An FDR-adjusted p value threshold of $q < 0.05$ was used to subset meaningful genes, which were used for pathway enrichment analysis.” (Line 505-507 in Page 21)

7a. In “Selection of orthologous genes” – it is not entirely clear how do you select your orthologs: do you use ENSEMBL orthology annotations between human and pig? If so, do you only use one-to-one orthologs?

Reply: In the revised version, we added more sentences to detail how we selected orthologous genes in the method section (Line 573-578 in Page 24): “Gene homology search was performed, using ensemble multiple species comparison tool (<http://www.ensembl.org/biomart/>) to generate annotations for human-porcine orthologs. We first downloaded a list of orthologous genes from the ENSEMBL BioMart (release 104). We removed any duplicate annotations as well as any orthologs that were not strictly 1:1. A high-quality orthologous gene list was extracted with both gene order conservation score and whole genome alignment score above 75 resulting in 12291 genes”.

7b. If you use one-to-many / many-to-many orthologs as well, how do you treat the “many”? e.g. -do you sum their expression together?

Reply: We only used one-to-one orthologs as mentioned in Line 575-576 in Page 24.

7c. Which ENSEMBL biomart version did you use?

Reply: We used ENSEMBL BioMart (release 104) as mentioned in Line 575 in Page 24.

7d. What do you mean by “whole genome alignment score above 75”? the reference given (#56) is probably the wrong reference as it seems unrelated. By “whole genome alignment” do you mean the entire gene body or just the CDS? Or the transcripts?

Reply: We selected the scoring criteria based on this reference. This reference was removed in the revision. ENSEMBL lists two methods for orthology quality-controls, one of which is whole genome alignment score. There is a detailed description for calculation of whole genome alignment score at https://uswest.ensembl.org/info/genome/compara/Ortholog_gc_manual.html#wga. We mentioned this in Line 578-579 in Page 24. Basically, whole genome alignment score is calculated based on the entire gene body.

8. In Supplementary Method 4 – cell suspension preparation – have the authors tried or considered different dissociation protocols? Why was this protocol used? Was there an attempt to enrich for a certain population of cells? What may be the biases and caveats in the protocol used? How many viable cells did you usually get following dissociation (based on either Trypan blue & microscopy and FACS)?

Reply: Our aim is to collect the enteric neuron and glia as many as we can from each tissue. Therefore, we chose the protocol summarized in the Supplementary Method 4. In the revised version, we added two sentences to address why this protocol was used: “Cell suspension preparation was performed according to the protocol presented by Smith et al.⁴ with modifications. The original protocol aims to isolate a mixed population of neurons and glia from the myenteric plexus in mouse” (Line 42-44 in Suppl.). In order to minimize artificial transcriptional changes during tissue dissociation, we added ActD to our media. Our RNA-seq data also proved the conclusion (Supplementary Fig. 7). In addition, one sentence “Application of ActD could minimize artificial transcriptional changes during tissue dissociation and enable unbiased characterization of cell types and their acute activation” was added in this section (Line 51-52 in Suppl.). On average, FACS helped us to collect viable cells, accounting for around 80.4%. Before we loaded the cell suspensions on the 10x Genomics Chromium platform, we also used Countess II FL Automated Cell Counter to check cell viability. We selected the cell suspensions with >70% viability. Regarding this aspect, we added more sentences “On average, the viable cells were collected via FACS accounted for around 80.4%. Before

the cell suspensions were loaded on the 10x Genomics Chromium platform, Countess II FL Automated Cell Counter was used to select the cell suspensions with >70% viability.” in Supplementary Method 4 (Line 65-67).

9. *In Supplementary Method 5 – you should mention the specific batch correction method used (Seurat has several such methods). Similarly, in: The UMAP algorithm was used for dimensionality reduction.” Should include more details.*

Reply: Thanks for reviewer’s suggestion. We added the details for the specific batch correction method and UMAP in Supplementary Method 5: “The Seurat Integration first used canonical correlation analysis (CCA) to project the data into a subspace to identify correlations across datasets. The mutual nearest neighbors (MNNs) were then computed in the CCA subspace and served as “anchors” to correct the data.” (Line 79-81 in Suppl.) and “All the dimensions of the low dimension representation were used as input to generate the UMAP plots with default parameters.” (Line 83-85 in Suppl.).

10. *n Fig 4 – how many cells and from how many individuals appear in panels A and B? (please add number of cells and individuals to Fig Legend)*

In methods you write that for single-cell, you used 4 individuals and loaded ~10,000 single cells per cell suspension. Can the same figures, colored by individual, be shown in Supp as well?

Reply: To address reviewer’s questions, we added sentences “The distinct cell clusters were derived from 2091 cells from one p-pC individual and from 3918 cells from one p-dC individual. Joint visualization based on four individuals of p-pC and p-dC could be found in Supplementary Fig. 3.” in the Legend for Fig. 4 (Line 835-837 in Page 37). We also added “Joint visualization of single-cell RNA sequencing data from four individuals of porcine proximal colon (p-pC) (a) and distal colon (p-dC) (b), related to Figure 4” in the Legend for Supplementary Fig. 3 (Line 162-163 in Suppl.).

Minor comments:

1) In Page 6, In: “*The interactions were defined when the networks mediated by DEG involved in WikiPathways of interest and Biological Processes (BPs) shared BPs.*” I guess that by “interactions” you refer to “cross-species similarities”? please rephrase.

Reply: We rephrased the description and added more backgrounds: “As mentioned above, the enrichment maps of GOBP terms were first created using the DEG lists. The DEG involved in the WikiPathways and Biological Processes (BPs) were searched in the enrichment maps. The GOBP terms were shared between different pathways (i.e. there were the DEG involved in the same GOBP terms), which was defined as the interactions between the networks mediated by the DEG.” in Line 151-155 in Page 7.

And, later in text – what do you mean by “Innate similarities”?

Reply: We deleted “Innate” in the text (Line 157 in Page 7).

2) In page 24, In: “*Their orthologous genes were defined in the porcine colonic ENS and applied for genetic risk prediction of human diseases*” You probably mean “identified”

Reply: Thanks for reviewer’s correction. We replaced “defined” with “identified” in the text (Line 632 in Page 26).

3) In Discussion, In: “*The networks mediated by the human orthologous genes from comparison of p-pC-MG and p-dC-MG did not realize 100% coverage of the linkages derived from the DEG lists in*

comparison of h-aC-MG and h-dC-MG”What do you mean by “did not realize”?

Reply: Thanks reviewer for pointing out this confusion. We replaced “did not realize 100% coverage of the linkages” with “partially overlapped those” in the text (Line 312 in Page 13).

Reviewers' comments:

Reviewer #1 (Remarks to the Author):

Most of my comments have been adequately addressed. Two remain that actually are related; these are still a concern to me as it appears the methods are demonstrably incorrect, assuming I understand the descriptions correctly.

Original reviewer comment #3 (in part).... To identify enrichment of function within DEG, normally all DEG are used and the frequency of a GO term is tested against an appropriate background (one suggestion would be to use as background all genes detected in the samples under study, as this list comprises those genes that could have been predicted to be DE). The authors did not justify selecting genes within a pathway first, and thus it is hard to see how enrichment is appropriately calculated.

Author Reply: Essentially the authors reiterate their method. I believe I understood it originally and I don't see any details that change my concern.

New Comment: My point was two-fold; the second point was to ask what background was used and this was not addressed in the author response. They do not state what background was used, so I assume that they used the software default which is the entire genome. This is inappropriate for the reason stated above by myself and the authors of the tool used. In their 2019 paper (doi: 10.1093/nar/gkz369) creators of g:Profiler state:

"By default, g:GOST uses the set of all annotated protein-coding genes as a background. In some experiments however, only a subset of genes or proteins is measured. For example, the targeted sequencing of only disease specific genes would imply enrichment of that disease association. Statistically, for these cases it is recommended and sometimes necessary to use custom background information when calculating the statistical enrichment significance. The custom background should include a list of genes that were actually measured during the biological experiment, such as all genes in the sequencing panel. This option allows to calculate a more precise evaluation of functional enrichment. "

Thus using as background only the genes that were detected in the RNAseq would be the appropriate background as they were the "targeted panel of genes" that could be declared DE or not.

Regarding the first point, if they actually selected subsets of genes that are annotated in specific Wikipathways "of interest", this is also not appropriate since the hypergeometric distribution test uses the length of the DEG list in the statistical calculation. Thus using a selected set of genes from the full DEG list will be biased toward enrichment of that selected pathway "of interest" when used in g:COSt (or ClueGO or any package using the hypergeometric distribution test for that matter).

Original reviewer comment #6. These concerns are important since the number of differentially expressed genes that predict an enriched GO term is crucial for interpretation. It appears they used as few as three (or even one!) genes to declare a biological process pathway is significantly different between samples (Suppl. Note 1). This is weak evidence for such a declaration, especially since they use such pathways to interpret results for much of the rest of the paper. A table showing the number of genes predicting such enrichment at the GO term level would be useful for all comparisons. This clarification is most important because it is critical to the main conclusion of the first section as well as much of the rest of the paper.

Author Reply:

- As we mentioned previously, based on all DEG derived from each comparison, two methods were used (g:Profiler and ClueGO) to create and visualize the functionally grouped networks of the enriched GO terms (Biological Processes) (Line 611-614 in the method section, Page 25). This means there are no any biases to interpret the data.

New Comment: The last sentence is a naïve statement. While it is often useful to use different approaches to test results, the methods have to be different (either the assumptions or the

mathematical treatment) to add confidence the results are well-supported. Since the two software packages used identify enriched annotations within gene lists in exactly the same way (hypergeometric distribution test), these methods do not test for bias introduced by the method used. And it is my contention (comment #3) that the enrichment testing was done inappropriately.

Reviewer #2 (Remarks to the Author):

The authors have answered most of my comments.

I do, however, still have several concerns regarding clarity, missing details and ability to reproduce the analysis, as follows:

(1) New Fig 1 requires a legend which expands further than the current legend: "Experimental design in the study", e.g. – total number of cells obtained, sub-region used, human versus pig comparison.

Also, in the middle panel, at the bottom – you write "4 of each 4 males and 8 females" – can you clarify?

(2) Statistics and general details of single-cells:

a. I still don't know how many single-cells were mapped per individual, per region and in total. The authors response of "On average, the final number of the cells used for cluster identification in each sample was ~2500." - is unclear and I am not sure it is indeed referring to what I was asking. I suggest that these details will be written in Fig 1 Legend.

b. In Methods, there are still many missing details regarding the protocols used. The authors write: "The viable single cells were loaded on the 10x Genomics Chromium platform and sequenced on an Illumina NextSeq 500 Sequencer (75 cycles)." Which 10x protocol and kit did you use? How many reads per cell on average did you obtain? (please add to the start of Methods)

(3) I still think that the authors should supply an online notebook / important scripts in a depository such as GitHub, which they haven't, despite my previous request (comment #3). This is standard in the field and is used by other to rerun the analyses.

We thank the reviewers for their valuable inputs that have improved the presentation of the data and discussion. A point-by-point reply to the comments and related changes in the revised manuscript are outlined below.

Reviewer # 1

Comment 1. *My point was two-fold; the second point was to ask what background was used and this was not addressed in the author response. They do not state what background was used, so I assume that they used the software default which is the entire genome. This is inappropriate for the reason stated above by myself and the authors of the tool used. In their 2019 paper (doi:10.1093/nar/gkz369) creators of g:Profiler state: “By default, g:GOST uses the set of all annotated protein-coding genes as a background. In some experiments however, only a subset of genes or proteins is measured. For example, the targeted sequencing of only disease specific genes would imply enrichment of that disease association. Statistically, for these cases it is recommended and sometimes necessary to use custom background information when calculating the statistical enrichment significance. The custom background should include a list of genes that were actually measured during the biological experiment, such as all genes in the sequencing panel. This option allows to calculate a more precise evaluation of functional enrichment.” Thus using as background only the genes that were detected in the RNAseq would be the appropriate background as they were the “targeted panel of genes” that could be declared DE or not.*

Reply: Thanks for reviewer’s suggestion. We re-analyzed the data using the genes that were detected in the bulk RNA-seq as the customized background. The results were similar to those obtained using the entire genome as background. We updated the Figures 2d, 3, 6, 7, 8 and 9. In the method section, we added “using the genes that were detected in the bulk RNA-seq as the customized background. The DEG lists were imported into g:Profiler (<https://biit.cs.ut.ee/gprofiler/>) or ClueGO v.2.5.6” (Line 617-618, Page 26).

Comment 2. *Regarding the first point, if they actually selected subsets of genes that are annotated in specific Wikipathways “of interest”, this is also not appropriate since the hypergeometric distribution test uses the length of the DEG list in the statistical calculation. Thus using a selected set of genes from the full DEG list will be biased toward enrichment of that selected pathway “of interest” when used in g:COSt (or ClueGO or any package using the hypergeometric distribution test for that matter).*

Reply: We may need to mention again that g:Profiler or ClueGo generates a functionally grouped GO annotation network that reflects the relationships between the terms based on the similarity of their associated genes. The primary criterion of the network is based on a hypergeometric test that is two-sided with $p < 0.05$, Benjamini-Hochberg correction, and kappa score > 0.4 . The assessment of the gene importance in a gene network can be quantified through the interactions between the target and other gene products. In the context of GO networks, this can be assessed through the number and enrichment scores of associated GO terms involving the target genes. Our intention to search the DEGs involved in the WikiPathways in the GO term networks is to find the GO terms associated with the WikiPathways. If the number and enrichment scores of GO terms related to the DEGs are larger, we can clarify the higher importance of the WikiPathways. At some points, this also reflects the importance of the DEGs in the WikiPathways in the gene regulatory networks. Therefore, we do not count on whether the small number of DEGs can represent the certain pathways. We just focus on the enrichment of the GO terms associated with the DEGs involved in the WikiPathways. On the other hand, we noticed that the hypergeometric test

relies on the same length of DEG lists to get the accurate results. What we actually do is that for all DEGs involved in a wiki-pathway, we just search one DEG at a time and obtain a p-value. Then we treat the DEGs in the wiki-pathway as dependent events and combine each p-value derived from each DEG by model averaging. In this way, we use the same length of the DEG list in the statistical calculation and acquire the accurate enrichment scores. We updated our method and stated that each DEG from a WikiPathway-matching DEG list was searched in the grouped networks of BPs and all terms (BPs) containing the DEG were highlighted. After we obtained the significance (corrected p-value by Benjamini-Hochberg) of the highlighted terms related to each DEG, by treating the DEGs in the DEG list as dependent events, we combined the significance derived from each DEG by model averaging to compute the FDR p-value of the specified WikiPathway using R/harmonicmean package (Line 620-625, Page 26 in Method section).

Comment 3. *The last sentence is a naïve statement. While it is often useful to use different approaches to test results, the methods have to be different (either the assumptions or the mathematical treatment) to add confidence the results are well-supported. Since the two software packages used identify enriched annotations within gene lists in exactly the same way (hypergeometric distribution test), these methods do not test for bias introduced by the method used. And it is my contention (comment #3) that the enrichment testing was done inappropriately.*

Reply: As we mentioned previously, for all DEGs involved in a wiki-pathway, we just search one DEG at a time and obtain a p-value. Then we treat the DEGs in the wiki-pathway as dependent events and combine each p-value derived from each DEG by model averaging (Mean Maximum Likelihood). In this way, we use the same length of the DEG list in the statistical calculation. This means there are not any biases to interpret the data. We updated our method section to specify the issue (Line 620-625, Page 26 in Method section). For transparency, we submitted a supplementary file showing the number of differentially expressed genes involved in each wiki-pathway in all comparisons.

Reviewer #2

Comment 1. *New Fig 1 requires a legend which expands further than the current legend: “Experimental design in the study”, e.g. – total number of cells obtained, sub-region used, human versus pig comparison. Also, in the middle panel, at the bottom – you write “4 of each 4 males and 8 females” – can you clarify?*

Reply: Thanks reviewer for pointing out this insufficiency. We expanded the legend of Fig 1 and added more details such as total number of loaded and mapped single cells in each sub-region and cross-comparison of the transcriptional profiling in the corresponding regions between porcine and human colon. We mentioned “Approximately 1400/3000/5300 cells were loaded onto the 10X Genomics Chromium Controller machine, resulting in 735/2091/2965 mapped single cells in p-pC, while in p-tC and p-dC, approximately 2800/2200/6300 and 2400/5000/4920 cells were loaded, resulting in 1993/1903/3057 and 1381/3918/2378 mapped single cells, respectively” and “Gene homology search was performed to obtain the orthologous genes between human and pig. The high-quality orthologous differentially expressed genes (DEG) derived from comparisons of MG between analogous colonic regions across species were extracted and the coverage of their functional linkages in the gene networks was used to reflect functional similarities across

species". We changed "4 of each 4 males and 8 females" to "4 of each segment from 5 males and 7 females" in the Fig. 1 and clarified it in the legend.

Comment 2. Statistics and general details of single-cells:

a. I still don't know how many single-cells were mapped per individual, per region and in total. The authors response of "On average, the final number of the cells used for cluster identification in each sample was ~2500." - is unclear and I am not sure it is indeed referring to what I was asking. I suggest that these details will be written in Fig 1 Legend.

b. In Methods, there are still many missing details regarding the protocols used. The authors write: "The viable single cells were loaded on the 10x Genomics Chromium platform and sequenced on an Illumina NextSeq 500 Sequencer (75 cycles)." Which 10x protocol and kit did you use? How many reads per cell on average did you obtain? (please add to the start of Methods)

Reply: We added the number of mapped single cells in the Fig. 1 legend. We mentioned "Approximately 1400/3000/5300 cells were loaded onto the 10X Genomics Chromium Controller machine, resulting in 735/2091/2965 mapped single cells in p-pC, while in p-tC and p-dC, approximately 2800/2200/6300 and 2400/5000/4920 cells were loaded, resulting in 1993/1903/3057 and 1381/3918/2378 mapped single cells, respectively".

We added the details regarding the protocols used in the method section (Lines 513-521, Pages 21-22). We mentioned "The scRNA-seq libraries were prepared from individual cells using the 10X Genomics platform. The Chromium Single Cell 3' Library & Gel Bead Kit v2, Chromium Single Cell 3' Chip kit v2 and Chromium i7 Multiplex Kit were used according to the manufacturer's instructions. On average, approximately 3200 cells from p-pC, 3800 cells from p-tC and 4100 cells from p-dC were loaded on the 10X Genomics Chromium platform and sequenced on an Illumina NextSeq 500 Sequencer (75 cycles). Sequencing was performed in paired-end mode. Cell Ranger 3.1.0 count function was used to align and quantify the reads against Sus_scrofa genome assembly, created using the Cell Ranger 3.1.0 mkref function with default parameters. This yielded an average of 1930, 2318 and 2559 mapped single cells (49884, 38516 and 43122 mean reads per cell) in p-pC, p-tC and p-dC, respectively."

(3) I still think that the authors should supply an online notebook / important scripts in a depository such as GitHub, which they haven't, despite my previous request (comment #3). This is standard in the field and is used by other to rerun the analyses.

Reply: Thanks for reviewer's suggestions. This study did not create any codes while we just followed the R scripts in the public GitHub repositories for data analyses. Thus no custom codes could be provided in a public repository. We added a Supplementary Table 5 to summarize the links of the public GitHub repositories where R scripts were listed and used in our study. They can be also used to rerun the analyses. A sentence "R scripts used for data analyses in our study can be found in the public GitHub repositories (Supplementary Table 5)" was added in Method section (Line 635-636, Page 26).

REVIEWERS' COMMENTS:

Reviewer #1 (Remarks to the Author):

Thanks for the additional details and information, and corrected analyses. No further concerns.

Reviewer #2 (Remarks to the Author):

The authors have answered most of my concerns. I still have a major concern regarding their cell numbers:

When you mention the cell number and individuals you say:

(1)

4 of each segment from 5 males and 7 females

and (2)

Approximately 1400/3000/5300 cells were loaded onto the 10X Genomics Chromium Controller machine, resulting in 735/2091/2965 mapped single cells in p-pC, while in p-tC and p-dC, approximately 2800/2200/6300 and 2400/5000/4920 cells were loaded, resulting in 1993/1903/3057 and 1381/3918/2378 mapped single cells, respectively

Are these the total numbers of mapped cells? from all 7 females and 5 males? were they mixed together before loading?

These details should be stated clearly in the manuscript.

I would like to emphasize here that these numbers are relatively low for papers that are solely focused on single-cell. While papers have been published with similar numbers and biologically valid conclusions can be made out of them, this is still considered a rather low number for papers of this type.

Reviewer # 2

Comment 1: *When you mention the cell number and individuals you say: (1) 4 of each segment from 5 males and 7 females and (2) Approximately 1400/3000/5300 cells were loaded onto the 10X Genomics Chromium Controller machine, resulting in 735/2091/2965 mapped single cells in p-pC, while in p-tC and p-dC, approximately 2800/2200/6300 and 2400/5000/4920 cells were loaded, resulting in 1993/1903/3057 and 1381/3918/2378 mapped single cells, respectively Are these the total numbers of mapped cells? from all 7 females and 5 males? Were they mixed together before loading? These details should be stated clearly in the manuscript.*

Reply:

(1) We thank the reviewer for these comments and sorry for the confusion. 4 of each segment from 5 males and 7 females are obtained from human tissue samples which were used only for laser-capture microdissection (LCM) coupled with bulk RNA-seq analysis, not for single cell RNA-seq. The scRNA-seq was performed only on 3 naïve pigs (2 males and 1 female) and we prepared 9 cell suspensions (each p-pC, p-tC and p-dC of 3 naïve pigs), each of which was loaded on the 10X Genomics Chromium platform and sequenced on an Illumina NextSeq 500 Sequencer. To avoid confusion, we added more details in Method section and the Legend of Figure. 1.

(2) We fully agree with the reviewer that the cell numbers and individuals are relatively low in this study. This is largely due to the challenge to isolate these cells, in particular the neurons in the ganglia from pig colon and their very low populations. To describe “the cell number and individuals” more clearly, we mentioned in Method section (Lines 516-522, Pages 21-22) :“Approximately, an average of 3200 cells from each p-pC, 3800 cells from each p-tC and 4100 cells from each p-dC were loaded on the 10X Genomics Chromium platform and sequenced on an Illumina NextSeq 500 Sequencer (75 cycles), generating 9 scRNA-seq datasets. Sequencing was performed in paired-end mode. Cell Ranger 3.1.0 count function was used to align and quantify the reads against Sus_scrofa genome assembly, created using the Cell Ranger 3.1.0 mkref function with default parameters. This yielded an average of 1930, 2318 and 2559 mapped single cells (49884, 38516 and 43122 mean reads per cell) in each of p-pC, p-tC and p-dC, respectively.” In the Legend of Figure 1 (Lines 804-809, Page 32), we added “Additionally, single-cell RNA-seq (scRNA-seq) was performed on the muscularis externa containing myenteric ganglia in three naïve pigs (2 males and 1 female). For each pig, approximately, an average of 3200 cells (1400/3000/5300) from p-pC, 3800 cells (2800/2200/6300) from p-tC and 4100 cells (2400/5000/4920) from p-dC were loaded on the 10X Genomics Chromium platform, generating 9 scRNA-seq datasets. This yielded an average of 1930 (735/2091/2965), 2318 (1993/1903/3057) and 2559 (1381/3918/2378) mapped single cells in each of p-pC, p-tC and p-dC, respectively.”